



# Stable isotope ratios in seawater nitrate reflect the influence of Pacific water along the Northwest Atlantic margin

Owen A. Sherwood[1], Samuel H. Davin[2], Nadine Lehmann[3], Carolyn Buchwald[3], Evan N. Edinger[4],
Moritz F. Lehmann[5], Markus Kienast[3]

[1]Department of Earth and Environmental Sciences, Dalhousie University, Halifax, NS, B3H 4R2, Canada
[2]GEOTOP Research Centre, Université du Québec à Montréal, Montreal, QC, H3C 3P8, Canada
[3]Department of Oceanography, Dalhousie University, Halifax, NS, B3H 4R2, Canada
[4]Department of Geography, Department of Biology, and Department of Earth Sciences, Memorial University of Newfoundland, St. John's, NL, A1B 3X9, Canada
[5]Department of Environmental Sciences, University of Basel, Basel, 4056, Switzerland

*Correspondence to*: Owen A. Sherwood (owen.sherwood@dal.ca)

**Abstract.** The flow of Pacific water to the North Atlantic exerts a globally significant control on nutrient balances between the two ocean basins, and strongly influences biological productivity in the Northwest Atlantic. Nutrient ratios of nitrate (N) versus phosphate (P) have previously been used to complement salinity characteristics in tracing the distribution of Pacific water in the North Atlantic. We expand on this premise and demonstrate that the fraction of Pacific water as determined by N/P ratios can be quantitatively predicted from the isotopic composition of sub-euphotic nitrate in the Northwest Atlantic. Our linear model thus provides a critically important framework for interpreting $\delta^{15}N$ signatures incorporated into both modern marine biomass, as well as organic material in historical and paleoceanographic archives along the Northwest Atlantic margin.

## 1 Introduction

Pacific water from the Bering Strait constitutes a major fraction of the polar outflow to the Northwest Atlantic Ocean (McLaughlin et al., 1996; Jones et al., 2003; Aksenov et al., 2010). The Pacific water is concentrated in an upper 300m halocline layer formed by winter cooling and sea-ice formation in the Chukchi and Beaufort Seas (McLaughlin et al., 2004; Steele et al., 2004). This halocline layer propagates through the Canadian Arctic Archipelago, then through Baffin Bay, and into the Labrador Sea via Davis Strait (Tang et al., 2004). A smaller fraction of Pacific water flows north of Greenland and into the North Atlantic via Fram Strait (Sutherland et al., 2009). The net eastward flow is driven by sea-level differences between the Pacific and Atlantic Oceans and, by redistributing heat and freshwater, plays a major role in the global thermohaline circulation (Tang et al., 2004; Woodgate et al., 2012; Tremblay et al., 2015; Carmack et al., 2016).

The flow of Pacific water to the North Atlantic is also critical with regards to the redistribution of nutrients between the two oceans. Pacific water has relatively high concentrations of nitrate ($NO_3^-$), phosphate ($PO_4^{3-}$), and silicate ($Si(OH)_4$) as it upwells



onto the shallow Bering Shelf. The high nutrient concentrations support high productivity on the Bering and Chukchi shelves (Arrigo and van Dijken, 2011), which in turn fuels high rates of sedimentary denitrification both in the shelf regions and along the Bering continental slope (Devol et al., 1997; Lehmann et al., 2005, 2007; Chang and Devol, 2009; Granger et al., 2011; Brown et al., 2015). The resulting excess in $Si(OH)_4$ and $PO_4^{3-}$ (relative to $NO_3^-$) is a significant source of these nutrients to the Atlantic (Torres-Valdés et al., 2013). In particular, the excess $PO_4^{3-}$ supports $N_2$-fixation in the Atlantic, thereby helping

to balance the global oceanic nitrogen budget (Yamamoto-Kawai et al., 2006).

Pacific-derived nutrients also influence biological productivity along the Northwest Atlantic shelf complex. The $NO_3^-$ deficit in Pacific water sets an upper limit on productivity, which otherwise would be higher in the presence of more $NO_3^-$-enriched Atlantic water (Harrison and Li, 2008). The $Si(OH)_4$ and $PO_4^{3-}$ excess influences plankton community composition (Harrison et al., 2013; Fragoso et al., 2017). Interannual and decadal-scale variability in the circulation of Pacific water into the Northwest

Atlantic may help to explain recent observed changes in the magnitude and composition of primary productivity with potential bottom-up effects on ecosystem functioning (Drinkwater et al., 2003; Greene et al., 2013; Townsend et al., 2015).

Given its importance to downstream circulation, nutrient budgets, and productivity, it is useful to track the distribution of Pacific water using chemical tracers. Jones et al. (1998) characterized $NO_3^-$:$PO_4^{3-}$ relationships for "pure" Pacific and Atlantic endmember waters. They further demonstrated that the concentrations of $NO_3^-$ and $PO_4^{3-}$ in a water sample relative to the

endmember relationships may be used to quantify the contribution of Pacific water (i.e., "fraction Pacific water", or fPW). With this approach, the spatial and depth distributions of fPW were used to map the flow of Pacific water through the Arctic and North Atlantic Oceans (Jones et al., 1998, 2003). The same approach has also been used to deconvolute fluxes of freshwater originating from Pacific water from that of sea ice meltwater and meteoric water (Yamamoto-Kawai et al., 2008; Sutherland et al., 2009; Azetsu-Scott et al., 2012; Benetti et al., 2016). In another study, time-series nutrient data were used to track fPW

and thereby infer changes in circulation patterns over a thirty-year period in Disko Bay, Greenland (Hanson et al., 2012). The use of $NO_3^-$:$PO_4^{3-}$ as a proxy for Pacific water has, however, important limitations. For example, the approach requires an assumption of constant stoichiometry associated with the uptake and recycling of nutrients, which may not hold in all regions (Michel et al., 2002; Mills et al., 2015). Moreover, sensitivity to $NO_3^-$ source and sink processes such as $N_2$ fixation and denitrification may lead to an under-, or overestimation of fPW, respectively. Finally, seawater $NO_3^-$:$PO_4^{3-}$ ratios are not

preserved in organisms or sedimentary archives, thus limiting their use in paleoceanographic contexts.

The nitrogen ($^{15}N/^{14}N$) and oxygen ($^{18}O/^{16}O$) isotope ratios in $NO_3^-$ (expressed as $\delta^{15}N_{NO3}$ and $\delta^{18}O_{NO3}$) represent a complementary tool to trace the distribution and modification of Pacific water, possibly addressing shortcomings related to the use of stoichiometric nutrient tracers. Coupled N and O isotope ratios provide insights into the internal cycling of $NO_3^-$, as well as input and removal processes. The preferential reaction of the lighter $^{14}N$ and $^{16}O$ during both phytoplankton uptake and

denitrification results in an enrichment of $\delta^{15}N$ and $\delta^{18}O$ of the dissolved $NO_3^-$ pool with a ratio of ~1 (Casciotti et al., 2002; Granger et al., 2004, 2008; Sigman et al., 2005). Conversely, the recycling or regeneration of $NO_3^-$ via nitrification (the



oxidation of ammonium ($NH_4^+$) to nitrite ($NO_2^-$) and $NO_3^-$) leads to a decoupling of the N and O isotopic signature of $NO_3^-$ (Sigman et al., 2005; Lehmann et al., 2005; Granger and Wankel, 2016). The $\delta^{15}N$ of newly nitrified $NO_3^-$ depends on the N isotopic composition of its source substrate and hence mirrors the isotopic signature of the organic matter exported from the surface. In contrast, the $\delta^{18}O$ of newly nitrified $NO_3^-$ remains independent from its N source and approaches the $\delta^{18}O$ signature of seawater ($\delta^{18}O_{H2O}$ +1.1‰; Casciotti et al., 2008; Sigman et al., 2009; Buchwald et al., 2012). Finally, the $\delta^{15}N_{NO3}$ also is reflected in the $\delta^{15}N$ of organisms and sediments that are used for paleoceanographic reconstructions (Altabet and Francois, 1994; Robinson et al., 2012, and references therein).

The goal of this paper is to evaluate the use of $\delta^{15}N_{NO3}$ and $\delta^{18}O_{NO3}$ as a new chemical oceanographic proxy for tracking the distribution of Pacific water to the Northwest Atlantic. We present new data from Baffin Bay, Davis Strait, and the Labrador Sea, highlighting differences in $NO_3^-$ isotopic ratios among the different water masses found in those regions. We evaluate the preservation of $\delta^{15}N_{NO3}$ and $\delta^{18}O_{NO3}$ signatures during southward advection of Pacific water from the Arctic Archipelago to the Labrador Shelf. Finally, we present a linear relationship between $\delta^{15}N_{NO3}$ and fPW that, given specific caveats, may be used to calculate fPW based on measurements of $\delta^{15}N_{NO3}$ in seawater, or the $\delta^{15}N$ preserved in organic materials, including paleoceanographic archives.

## 2 Materials and Methods

### 2.1 Sample collection and nutrient measurements

Seawater samples were collected opportunistically during four different expeditions that sampled 25 stations along the NW Atlantic margin from the mid-Labrador Shelf to northern Baffin Bay between the years 2005 and 2016 (Fig. 1). New and previously published data are presented. New data were collected during 1) expedition MSM45 of the *Maria S. Merian* in August 2015 and 2) an ArcticNet expedition (AMD-2016-002a) of the Canadian Coast Guard Ship (CCGS) *Amundsen* in July through September 2016. Previously published data are from 3) expedition HUD-2005-016 of the CCGS *Hudson* in June 2005; 4) and a GEOTRACES (GN02) expedition aboard the CCGS *Amundsen* in July/August 2015. Stations associated with each expedition are indicated in the Fig. 1 stations legend. Sample collection and analytical protocols for the MSM45 and AMD-2016-002a expeditions are given below. Protocols for the HUD-2005-016 expedition are provided in Sherwood et al. (2011) and for the GEOTRACES expedition in Lehmann et al. (2019). Samples were collected under ice-free conditions during all expeditions. Station and bottle data are provided as a supplementary data file.

Samples from the MSM45 and AMD-2016-002a expeditions were collected with a rosette water sampler holding 24 x 10L Niskin bottles mounted with a conductivity-temperature-depth (CTD) profiler equipped with sensors for dissolved oxygen and fluorescence. During the MSM45 expedition, samples for nutrient and $NO_3^-$ isotope analysis were collected from the Niskin bottles into separate, triple-rinsed, 60 ml high-density polyethylene (HDPE) bottles with pre-filtration through 0.45 μm



**Figure 1: Map and inset detail of study region with World Ocean Atlas (WOA18) climatological mean temperature at 100 m water
depth (colour shading), bathymetry (black contours), major surface currents (arrows) and sampling stations. Abbreviations: West
Greenland Current (WGC), Baffin Island Current (BIC), Labrador Current (LC), Nares Strait (NS), Jones Sound (JS), Lancaster
Sound (LS), Hecla and Fury Straits (HFS), Hudson Strait (HS), Hatton Basin (HBn), Saglek Bank (SB), Hamilton Bank (HB).**

surfactant-free cellulose acetate (SFCA) membrane filters and stored at -20 °C. Concentrations of $NO_3^-$, $NH_4^+$, and $Si(OH)_4$
were measured post-cruise at Dalhousie University according to standard protocols (Grasshoff, 1969) using a Bran and Luebbe
autoanalyzer III. During the AMD-2016-002a expedition, samples for nutrient analysis were pre-filtered (0.2 μm) into 15 ml
acid-rinsed centrifuge tubes. The samples were analyzed onboard the ship for $NO_3^-$, $NO_2^-$, $PO_4^{3-}$, and $Si(OH)_4$ concentrations
using a Bran and Luebbe autoanalyzer III following standard protocols (Grasshoff, 1969). Samples for $NO_3^-$ isotope analysis
were collected into acid-cleaned and triple-rinsed 60 ml HDPE bottles without pre-filtration, and stored at -20 °C.





### 2.2 NO$_3^-$ isotope analyses

Seawater samples were prepared for the measurement of dual N and O isotope ratios in NO$_3^-$ following the denitrifier method (Sigman et al., 2001; Casciotti et al., 2002). This method quantitatively converts NO$_3^-$ present in the water samples to nitrous oxide (N$_2$O) by introducing cultured denitrifying bacteria (*Pseudomonas chlororaphis* f. sp. *aureofaciens,* ATCC# 13985) that lack N$_2$O reductase activity. The resulting N$_2$O gas is then analyzed by isotope ratio mass spectrometry (IRMS). Isotopic ratios are reported in delta notation following Eq. (1):

$$\delta^{15}N \text{ or } \delta^{18}O = [R_{sample}/R_{standard} - 1] \times 1000, \tag{1}$$

where R represents either $^{15}N/^{14}N$ or $^{18}O/^{16}O$, the standard is the N$_2$ in the atmosphere (air) or the oxygen in Vienna Standard Mean Ocean Water (VSMOW), and the units are reported as per mille (‰) deviation from the standard ratios.

Samples from the MSM45 expedition were analyzed at Dalhousie University using a Thermo Fisher Delta V Plus IRMS interfaced with a Thermo Gas Bench inlet. Data were calibrated using seawater-based reference material USGS-32 ($\delta^{15}N$ = +180 ‰ vs. air, $\delta^{18}O$ = +25.7 ‰ vs. VSMOW), USGS-34 ($\delta^{15}N$ = -1.8 ‰ vs. air, $\delta^{18}O$ = -27.9 ‰ vs. VSMOW) and IAEA-N3 ($\delta^{15}N$ = +4.7 ‰ vs. air, $\delta^{18}O$ = +25.6 ‰ vs. VSMOW) (Böhlke et al., 2003; Gonfiantini, 1984). NO$_2^-$ concentrations were always < 0.2 µM, and no prior NO$_2^-$ removal was performed. Analytical reproducibility based on replicate measurements averaged 0.2 ‰ for $\delta^{15}N$ and 0.4 ‰ for $\delta^{18}O$.

Samples from the AMD-2016-002a expedition were analyzed at the University of Basel using a ThermoFisher Delta V Plus IRMS with a customized purge and trap system (modified after McIlvin and Casciotti, 2010, 2011). NO$_3^-$ samples were analyzed in parallel to isotope reference material USGS-34 and IAEA-N3. NO$_2^-$ concentrations were always < 0.4 µM, and no prior NO$_2^-$ removal was performed. Analytical reproducibility based on replicate measurements averaged 0.2 ‰ for $\delta^{15}N$ and 0.3 ‰ for $\delta^{18}O$.

### 2.3 Definitions and Calculations

Seawater potential temperature ($\theta$) and potential density anomaly referenced to surface pressure ($\sigma_\theta$) were calculated from CTD data using the "oce" package in the R computing platform (Kelley and Richards, 2017). Water masses (Table 1) were operationally defined on the basis of $\theta$ and/or $\sigma_\theta$ thresholds following published conventions (Stramma et al., 2004; Fratantoni and Pickart, 2007; Azetsu-Scott et al., 2012). Note that Halocline Water (HW) and Labrador Shelf Water (LShW) are distinct water masses, despite overlapping $\theta$ and $\sigma_\theta$ characteristics. The latter is formed in the Hudson Strait area from mixing of HW with Irminger Water (IW) and Hudson Bay outflow water (Sutcliffe et al., 1983; Straneo and Saucier, 2008). The base of the biologically productive zone ($z_P$) was determined from CTD profiles of chlorophyll fluorescence as the shallowest depth below the subsurface chlorophyll maximum where values < 0.1 mg/m$^3$ were encountered. Apparent Oxygen Utilization (AOU) was



calculated from CTD *in situ* dissolved oxygen profiles using equations in Weiss et al. (1970). The N* parameter, quantifying $NO_3^-$ to $PO_4^{3-}$ imbalances relative to Redfield stoichiometry, was calculated following Eq. (2):

$$N^* = NO_3^- - 16 PO_4^{3-} + 2.95, \tag{2}$$

where the constant 2.95 forces a global mean N* of zero (Gruber and Sarmiento, 1997; Deutsch et al., 2001). Regenerated $PO_4$

($P_{reg}$) was calculated from AOU and the stoichiometric constant of Anderson and Sarmiento (1994) from Eq. (3):

$$P_{reg} = AOU/170. \tag{3}$$

The ratio of regenerated to total $PO_4$ is reported as $P_{reg/tot}$.

Fraction Pacific water (fPW) was calculated from N and P concentration data in relation to N:P relationships for Pacific and

Atlantic endmembers, following Jones et al. (1998). For the purposes of data representativity, accessibility, and propagation of error calculations, we derived new equations for Atlantic and Pacific waters using public domain data from the 2019 version of the Global Ocean Data Analysis Project (GLODAP; Olsen et al., 2020). The N:P relationships are sensitive to the choice of dissolved inorganic nitrogen (DIN) species ($NO_3^-$, $NO_2^-$, $NH_4^+$), particularly in highly productive shelf regions where a significant fraction of the total dissolved N may not be fully nitrified (Yamamoto-Kawai et al., 2008; Mills et al., 2015). The

equations reported here use only $NO_3^-$ concentrations, as the inclusion of $NO_2^-$ and $NH_4^+$ was found to have a negligible impact on fPW calculations (less than calculated uncertainties, see below).

Pacific endmember data were selected from a region encompassing the Canada Basin of the Beaufort Sea (Fig. S1a). The data were filtered (bottom depths > 500m), to exclude shelf waters where $NH_4^+$ accounts for a significant fraction of the DIN. Data

were further filtered ($S \leq 33.5$; $NO_3^- > 2.5 \, \mu M$) to exclude waters of Atlantic origin, as well as data from below a kink in the $NO_3^-$ vs. $PO_4^{3-}$ relationship at low nutrient concentrations (Yamamoto-Kawai et al., 2008). The resulting relationship for Pacific water (PW) was calculated following Eq. (4):

$$NO_3^{PW} = 14.07 \pm 0.09 \times PO_4^{PW} - 11.53 \pm 0.15, (r^2 = 0.85, n = 4109). \tag{4}$$

This relationship is within error of the one reported in Yamamoto-Kawai et al. (2008), which was based on total DIN vs. $PO_4^{3-}$

data for a region also encompassing the Chukchi Sea (Fig. S1b). This indicates that the $NH_4^+$ that accumulates on the Bering and Chukchi shelves in summer is largely nitrified by the time the Pacific-origin shelf waters reach the Canada Basin and Amundsen Gulf (Brown et al. 2015; Granger et al., 2018). In other words, the use of total DIN (instead of just $NO_3^-$) to define the Pacific N:P relationship, would have a negligible effect on the derived Pacific endmember relationship in Eq. (4).

Atlantic endmember data were obtained from the Irminger Sea (Fig. S1a), which, based on drifter trajectories, represents the source region for waters entering the Labrador Sea via the Irminger Current (Cuny et al., 2002; Jakobsen et al., 2003). Data from the region were filtered ($S \geq 35$, $NO_3^- > 2.5 \, \mu M$) to exclude polar waters entering the area through Fram Strait (Sutherland et al., 2009), and waters affected by nutrient drawdown. The resulting relationship for Atlantic water (AW) entering the Labrador Sea was calculated following Eq. (5):



$NO_3^{AW} = 15.54 \pm 0.10 \times PO_4^{AW} - 0.26 \pm 0.10$, ($r^2 = 0.89$, n = 2669). (5)

For any given sample $PO_4^{3-}$ concentration, Eq. (4) and (5) define theoretical endmember $NO_3^{PW}$ and $NO_3^{AW}$ concentrations, respectively. The fPW was then calculated from the sample $NO_3^-$ concentration in relation to $NO_3^{PW}$ and $NO_3^{AW}$, following Eq. (6):

$fPW = (NO_3^{sample} - NO_3^{AW}) / (NO_3^{PW} - NO_3^{AW})$. (6)

Negative values were considered to be devoid of Pacific water and were set to zero. Analytical error in $NO_3^-$ and $PO_4^{3-}$ measurements averaged 1 % and 2 %, respectively. Errors propagated through Eq. (4), (5), and (6) with statistical bootstrapping (n = 10,000, with uniform distributions) resulted in uncertainties of ±3 % (at the 95 % confidence level) on fPW estimates.

**Table 1: Water mass definitions and statistical summaries (mean ± 1 st. dev.) of physical and chemical properties by water mass.**
**HW: Halocline Water; BBW: Baffin Bay Water; LShW: Labrador Shelf Water; IW: Irminger Water; LSW: Labrador Sea Water; NEADW: Northeast Atlantic Deep Water; DSOW: Denmark Strait Overflow Water. All properties were calculated for waters below the biologically productive zone ($z_p$).**

| Water Mass | Operational Definition | Depth (m) | S | $\theta$ (°C) | AOU (μM) | $NO_3^-$ (μM) | $PO_4^{3-}$ (μM) | $Si(OH)_4$ (μM) | $\delta^{15}N_{NO3}$ (μM) | $\delta^{18}O_{NO3}$ (μM) | N* (μM) | $P_{reg/tot}$ | fPW |
|---|---|---|---|---|---|---|---|---|---|---|---|---|---|
| HW | $\sigma_\theta \leq 27.3$, $\theta \leq 0$ | 149 ± 53 | 33.30 ± 0.45 | -1.13 ± 0.50 | 60 ± 22 | 10.1 ± 2.0 | 0.98 ± 0.10 | 12.6 ± 3.0 | 6.0 ± 0.5 | 1.3 ± 1.2 | -2.6 ± 1.6 | 0.36 ± 0.12 | 0.39 ± 0.13 |
| BBW | $27.5 > \sigma_\theta \leq 27.8$, $\theta \leq 2$ | 900 ± 507 | 34.49 ± 0.04 | 1.06 ± 0.75 | 143 ± 49 | 18.5 ± 2.6 | 1.40 ± 0.26 | 41.6 ± 25.5 | 6.1 ± 0.5 | 1.1 ± 0.2 | -0.7 ± 1.7 | 0.58 ± 0.10 | NA[a] |
| LShW | $\sigma_\theta \leq 27.4$, $\theta \leq 2$ | 109 ± 39 | 33.63 ± 0.46 | 0.10 ± 0.97 | 44 ± 8 | 10.2 ± 1.8 | 0.88 ± 0.06 | 8.8 ± 2.0 | 5.4 ± 0.7 | 2.1 ± 0.5 | -0.4 ± 1.6 | 0.29 ± 0.04 | 0.22 ± 0.12 |
| IW | $27.3 > \sigma_\theta \leq 27.68$, $\theta > 2$ | 270 ± 140 | 34.61 ± 0.19 | 3.20 ± 0.80 | 52 ± 19 | 14.5 ± 1.6 | 1.00 ± 0.07 | 10.2 ± 3.8 | 4.9 ± 0.3 | 2.1 ± 0.5 | 1.9 ± 0.8 | 0.31 ± 0.11 | 0.04 ± 0.05 |
| LSW | $27.68 > \sigma_\theta \leq 27.80$, $\theta > 2$ | 893 ± 438 | 34.88 ± 0.03 | 3.54 ± 0.25 | 36 ± 11 | 16.1 ± 0.8 | 1.02 ± 0.05 | 8.5 ± 1.8 | 4.8 ± 0.3 | 2.1 ± 0.4 | 2.6 ± 0.4 | 0.18 ± 0.05 | 0.00 ± 0.02 |
| NEADW | $27.80 > \sigma_\theta > 27.88$ | 2245 ± 255 | 34.92 ± 0.00 | 2.83 ± 0.28 | 51 ± 5 | 16.0 ± 0.5 | 1.04 ± 0.02 | 10.8 ± 0.8 | 4.7 ± 0.3 | 1.8 ± 0.3 | 2.3 ± 0.1 | 0.28 ± 0.00 | 0.00 ± 0.00 |
| DSOW | $27.88 > \sigma_\theta$ | 2837 ± 151 | 34.91 ± 0.01 | 1.95 ± 0.26 | 53 ± 6 | 15.2 ± 0.4 | 1.01 ± 0.01 | 12.2 ± 1.3 | 4.9 ± 0.2 | 2.0 ± 0.3 | 2.0 ± 0.1 | NA | 0.02 ± 0.01 |

[a]It is not possible to calculate fPW for BBW. See section 3.2.4 for details

# 3 Results and Discussion

## 3.1 Hydrographic summary

The 25 stations that were sampled for this study are grouped into five hydrographic regimes on the basis of common water column properties. These regimes are distributed along the net transport pathway of Pacific water and are therefore ideally situated for investigating the distribution of $NO_3^-$ isotopic ratios with respect to fPW. A summary of hydrographic properties





**Figure 2: Temperature - salinity diagram for individual stations. For each profile, the 30m depth level is indicated by open symbols; the bottom level is indicated by shaded symbols. Data from < 30 m omitted for clarity. Bold black lines indicate temperature and σ_T limits for water masses discussed in text:  HW, Halocline Water, LShW, Labrador Shelf Water; BBW, Baffin Bay Water; IW, Irminger Water; LSW, Labrador Sea Water; NEADW, Northeast Atlantic Deep Water; DSOW, Denmark Strait Overflow Water.**






by regime follows below. To help visualize the data, the data are colour-coded by hydrographic regime consistently throughout the subsequent figures. A diagram of θ-S data is shown in Fig. 2, with delineations for the different water masses. Depth profiles are shown for all stations in Fig. 3 and separately for each hydrographic regime in Fig. S1-S5.


The Baffin Bay regime was represented by three stations (BB2, BB3, CAA3). Station BB2 was located inside the central Baffin gyre at a depth of 2300 m. BB3 was located along the path of the Baffin Island Current at a depth of 1243 m. CAA3 was located at the southern side of Lancaster Sound at a depth of 690 m. The three stations displayed similar θ and S profiles, characteristic of Baffin Bay more broadly (Fig. 3; Tang et al., 2004). A surface layer, formed by summer warming and melting,

extended down to 20-30 m. Below this, a layer of almost isothermal, cold (θ < -1.5 °C) water, increasing in salinity from < 32 to 34, extended down to 200 m. This layer represents the Pacific-sourced HW, which is formed in the Beaufort Sea and adjacent shelves, and then modified by regional winter cooling and sea ice formation in northern Baffin Bay and along the northwestern Greenland coast (Bourke et al., 1989; Münchow et al., 2015; Rysgaard et al. 2020). Below the HW, a warmer (θ approaching +2 °C) and saltier (S > 34) layer extended down to 700 m. This layer represents the diluted remnants of Atlantic-sourced IW,

often referred to as West Greenland Intermediate Water, which flows northward via the West Greenland Current and spreads throughout the entire Baffin Bay (Tang et al., 2004; Münchow et al., 2015). The θ and S data over this depth interval at CAA3 were more variable, reflecting the interleaving of different water masses by the complex tidal currents in Lancaster Sound (Fig. S3; Prinsenberg and Hamilton, 2005). At stations BB2 and BB3, the waters below 700 m form a distinct tail on the θ-S diagram (Fig. 2). We refer to this as "Baffin Bay Water" (BBW; θ < 2 °C, 27.5 > σθ ≤ 27.8), which, for convenience, groups

waters generally referred to as Baffin Bay Deep Water for 1200 < z < 1800 m and Baffin Bay Bottom Water for z > 1800 m (Tang et al., 2004), as well as the shallower waters from 700 < z < 1200 m. The BBW is also distinguished by the rapid increase in AOU with depth (Fig. 3c).

The northern Davis Strait regime was represented by four stations (177, 179, BB1, ROV7). Station 177 was located within 2

km of coastal Baffin Island at a depth of 376 m. Despite the coastal location of station 177, it is hydrographically connected to more open water via a deep, northeast-trending cross-shelf trough (Broughton Trough). Station 179 was located on the Baffin shelf break at 186 m. Station BB1 was located on the northern flank of the Davis Strait sill at a depth of 1042 m. Station ROV7 was located over the Greenland slope (Disko Fan) at a depth of 932 m. Hydrographic profiles at these four stations were similar to those of the Baffin Bay regime, with the characteristic HW and IW layers (Fig. 3). A seemingly thicker surface layer

extending down to > 50m at station 177 is a result of the later sampling date (late September) than at the other three stations, which were sampled in late July/early August. The HW layer was thicker at stations 177 and 179, which are located in the path of the Baffin Island Current and thins out toward the more centrally located BB1 and ROV7, also evident from the shallowing isopycnals (Fig. S4 σθ profiles; Tang et al., 2004; Azetsu-Scott et al., 2012). Stations BB1 and ROV7 sampled the IW (θ > 2 °C, S > 34.4) from 300-500 m, and BBW below about 700 m.






Figure 3: Depth profiles for (a) potential temperature, (b) salinity, (c) apparent oxygen utilization (AOU), (d) PO₄³⁻, (e) NO₃⁻, (f) N*, (g) $\delta^{15}$N of NO₃⁻, and (h) $\delta^{18}$O of NO₃⁻.



The Labrador Shelf regime comprised seven stations (009, 018, 024, 030, 154, 147, 143). Station 009 was located over a >
900 m deep basin in the main channel of the Hudson Strait but had a similar hydrographic profile to the other stations on the
Labrador Shelf. Stations 018, 024, and 030 were located on an along-shelf transect, located at depths of 200 m, 534 m, and
535 m, respectively. Stations 154, 147, and 143 were located along an outer cross-shelf transect of Hamilton Bank at depths
of 202 m, 245 m, and 344 m, respectively. A surface layer extended down to about 30 m at all stations, underlain by the
remnants of the HW, modified by tidal mixing and warming southward of Davis Strait (Fig. 3; Tang et al., 2004). This layer
is often called the "Cold Intermediate Layer" (Colbourne et al., 2016), but herein is referred to as "Labrador Shelf Water"
(LShW; $\theta < 2$ °C, S < 34.2) for ease of reference. The LShW extended down to between 150-300 m and was underlain by IW
where the bottom depth exceeded 300 m. The influence of IW increased from west to east, as becomes apparent from the cross-
shelf increase in $\theta$ and S at stations 030, 154, 147, and 143 (Fig. S5; Fratantoni and Pickart, 2007).


The outer Hudson Strait regime comprised five stations (ROV1, ROV2, ROV3, ROV5, ROV6) concentrated around an area
seaward of the Hudson Strait, around the shelf break (Fig. 1 inset). Stations ROV1 and ROV5 were located on the sill of an
outer shelf bathymetric depression (Hatton Basin) at approximately 500 m water depth. Stations ROV2 and ROV3 were located
along the northern flank of Saglek Bank at 279 and 436 m, respectively. Station ROV6 was located further north, at 456 m
depth, but had similar hydrography as the other four stations (Fig. 3). The surface and bottom currents in these areas are quite
strong, up to 0.60 m s$^{-1}$ at station ROV3, generally flowing NW to SE, but with a strong tidal influence linked to the macrotidal
oscillation in Frobisher Bay (Zedel et al., unpublished bottom current meter data from NE Saglek Bank). The surface layer
extended down to about 30 m at all stations. Nutrient data, specifically PO$_4^{3-}$ concentrations and N* values, discussed below,
clearly distinguished the surface waters of this regime from the other regimes presented above. The water mass structure was
overall similar to that of the Labrador Shelf regime, but with a thinner, warmer and saltier layer of LShW underlain by warmer
and saltier IW (Fig. S6).

Finally, the Labrador Basin regime comprised six stations (006, 013, 016, 033, LS2, K1) located in the deep waters of the
Labrador Sea, at depths of 1280 m (station 013) to 3292 m (station 033). Hydrographic profiles at these stations (Fig. 3) reflect
the well-known water mass structure in the Labrador Sea (e.g. Yashayaev and Loder, 2016). Doming of isopycnals leads to
the thinning and shoaling of the IW layer from the margins (e.g. Station 013, IW: 70-500 m) to the center of the basin (e.g.
Station K1, IW: 30-150 m) (Fig. S7). Below the IW, a thick layer of Labrador Sea Water (LSW: $27.68 > \sigma_\theta \leq 27.80$, $\theta > 2$
°C), extended down to 1500-2000 m, underlain by Northeast Atlantic Deep Water (NEADW: $27.80 > \sigma_\theta > 27.88$) to 2400-
2700 m, then Denmark Strait Overflow Water (DSOW: $\sigma_\theta > 27.88$).





## 3.2 Nutrients

### 3.2.1 Near-surface nutrients

Sampling was conducted in the months of June – August, which follows the spring bloom throughout most of the study region (Tremblay et al., 2006; Frajka-Williams et al., 2010). Complete or near-complete utilization of $NO_3^-$, $PO_4^{3-}$, and $Si(OH)_4$ was observed in the upper 30 m of the water column at all sites, with evidence of partial nutrient utilization to < 120 m. Minima in $NO_3^-$ in the surface waters averaged < 1 µM and did not vary by hydrographic region (Fig. 3e). Minima in $PO_4^{3-}$, by contrast, exhibited a striking bimodal distribution with respect to region, with concentrations < 0.1 µM for most of the outer Hudson Strait and Labrador Basin stations, and > 0.4 µM for all of the Baffin Bay, Davis Strait, and Labrador Shelf stations (Fig. 3d). Minima in $Si(OH)_4$ exhibited a similar bimodality (< 1 µM and > 5 µM) with respect to hydrographic regions (Fig. S1-S5). Thus, $NO_3^-$ was relatively more limiting to primary production than either $PO_4^{3-}$ or $Si(OH)_4$ in the colder/fresher hydrographic regions, as observed previously (Tremblay et al., 2006; Harrison and Li, 2008; Martin et al., 2010; Ferland et al., 2011; Fragoso et al., 2017).

### 3.2.2 Sub-surface nutrients

Nutrient concentrations generally stabilized below the biologically productive zone (depth > $z_P$), with the exception of BBW, in which concentrations increased rapidly with depth (Fig. 3d, e). The elevated concentrations result from in situ nutrient regeneration in Baffin Bay Deep and Bottom Water (Jones et al., 1984; Tremblay et al., 2002; Lehmann et al. 2019). Baffin Bay is a 2300 m deep basin enclosed by < 700 m deep sills. The enclosed bathymetry and permanent halocline restrict circulation, thereby trapping particulate organic matter (POM) and remineralized nutrients. More precisely, given the long residence time of the deep and bottom waters (77-1450 years; Top et al., 1980; Wallace et al., 1985), high fluxes of POM originating from the productive northern Baffin Bay (Klein et al., 2002; Tremblay et al., 2002; Lalande et al., 2009) accumulate at depth. The subsequent *in situ* remineralization of this sinking POM leads to the observed increase in nutrients, seen as an increase in $P_{reg/tot}$, and drawdown of oxygen (increase in AOU) in the deep basin (Fig. 3c-e). While $O_2$ concentrations remain too high to support denitrification in the water column, dissimilatory $NO_3^-$ consumption in the sediments is supported by the low oxygen concentrations in the water above, and acts as a potential sink for dissolved $NO_3^-$ in the lower water column (Lehmann et al., 2019). Indeed, BBW had two-fold higher $PO_4^{3-}$ ($1.4 \pm 0.3$ µM) and three-fold higher $Si(OH)_4$ concentrations ($41 \pm 25$ µM) than any of the other water masses (Table 1), but only somewhat higher $NO_3^-$ ($19 \pm 3$ µM). For water masses other than BBW, there were significant differences in $NO_3^-$ and $PO_4^{3-}$, but not $Si(OH)_4$ (1-way ANOVA). HW and LShW had lower $NO_3^-$ concentrations (~10 µM) than IW, LSW, NEADW (~15 µM). The distribution of $PO_4^{3-}$ by water mass was similar, except that the concentration in HW (0.98 µM) was closer to that of IW, LSW, and NEADW (> 1 µM) than LShW (0.88 µM).





**Figure 4: NO₃⁻ vs. PO₄³⁻ data for samples from all water depths in this study, with lines representing empirically-derived Atlantic and Pacific water endmember N:P relationships. Endmember lines are enclosed by 95 % confidence intervals. Coloured red to blue lines between Atlantic and Pacific endmembers represent lines of constant fraction Pacific water (fPW), in increments of 0.2. Also shown are grey dashed lines of constant N\*.**

### 3.2.3 Nutrient ratios

$NO_3^-$ to $PO_4^{3-}$ stoichiometry is expressed in profiles of N\*, which were available for 19 of the 25 stations with paired $NO_3^-$ and $PO_4^{3-}$ concentration data (Fig. 3f). Positive N\* reflects a water mass history of excess fixed $NO_3^-$, e.g., by net $N_2$-fixation; negative N\* reflects a $NO_3^-$ deficit, relative to the mean global ocean (Gruber and Sarmiento, 1997), induced by denitrification in the broadest sense (i.e., including other modes of suboxic DIN transformations to N such as anammox). N\* signatures can be imported from other ocean regions, or can be generated within a given water mass or region, depending on biogeochemical conditions. Positive N\* occurred throughout most of the outer Hudson Strait and Labrador Basin profiles. The water masses



LSW, NEADW, and DSOW (z > 200 m) all showed mean N* > 2 µM (Table 1), consistent with an Atlantic origin (Gruber and Sarmiento, 1997; Jenkins et al., 2015). The deflections to lower N* at z ~ 100 m (Fig. 3f) correspond to IW, which has lower N* due to mixing at the shelf-slope front (Cuny et al., 2002; Fratantoni and Pickart, 2007). Negative N* occurred through most of the Baffin Bay, Davis Strait, and Labrador Shelf profiles (Fig. 3f). As noted earlier, HW partially originates from the

Bering and Chukchi shelf areas, where sedimentary denitrification fueled by high water column productivity acts as a sink for dissolved $NO_3^-$. The resulting pronounced minimum in N* (< -10 µM; Yamamoto-Kawai et al., 2008; Mills et al., 2015) propagates via HW through the Canadian Arctic Archipelago and into Baffin Bay (Carmack and McLaughlin, 2011; Tremblay et al., 2015). This import of fixed-N deficient waters explains the lowest N* values in HW (Table 1), with minima < 6 µM at stations CAA3, BB3, BB2, BB1, 177, and 179 (Fig. 3f). BBW had the next most negative N*, likely due to the upward

propagation of partially denitrified bottom water nutrients (Tremblay et al., 2002; Lehmann et al., 2019). LShW had the most variable N* signatures, resulting from the mixing of HW and IW. The effect of this mixing is clearly evident in the cross-shelf increase in N* on the Labrador Shelf, from < -3 µM at station 154 to > 2 µM at station 143 (Fig. S5).

### 3.2.4 Fraction Pacific water

Figure 4 shows $NO_3^-$ and $PO_4^{3-}$ data plotted with lines of constant fPW. For comparison purposes, the figure also shows lines

of constant N*. The Atlantic endmember line (fPW = 0) coincides with the line of N* = 2 µM (Fig. 4), which is also the average N* value for North Atlantic intermediate waters (Gruber and Sarmiento, 1997). The Pacific line (fPW = 1) falls between the lines where N* = -10 to -12 µM. Thus, depth profiles of fPW mirror those of N* (Fig. S3-S7). Data from the Labrador Basin regime fall on or close to the Atlantic line because the water masses at these stations (IW, LSW, NEADW, DSOW) are mostly Atlantic-sourced. Data from the other regimes plot increasingly towards the Pacific line in the order of

Hudson Strait, Labrador Shelf, Davis Strait, and Baffin Bay. Maxima in fPW (> 0.6) were found within the core of HW sampled at Lancaster Sound (CAA3) and along the path of the Baffin Island Current (BB2, BB3, BB1, 177 179) (Fig. S3, S4).

In Fig. 4, the group of values with $PO_4^{3-}$ > 1.25 µM and $NO_3^-$ > 17.5 µM corresponds to BBW. The data fall along a N:P trajectory with a slope of 9.6 +/- 0.3. This slope is considerably lower than the slopes of either Atlantic or Pacific endmember

waters. It arises from *in situ* remineralization of POM, as indicated by $P_{reg/tot}$ values > 0.5 (Fig. S3, S4), with a partial loss of $NO_3^-$ via sedimentary denitrification (Lehmann et al., 2019). The denitrification generates N* values as low as -4.3 µM. Thus, the process that leads to low N* in BBW is separate and distinct from the processes that generate low N* in HW. As a result, it is not possible to calculate fPW for BBW, because remineralization and denitrification overprint the pre-formed N:P signatures (Jones et al., 2003).


Another complication with fPW estimates, as noted in section 2.3, is that elevated concentrations of $NO_2^-$ and $NH_4^+$ may alter apparent N:P ratios with respect to the derived endmember relationships. Within the overall study region, $NO_2^-$ and $NH_4^+$ concentrations below the euphotic zone are generally < 1 µM (Harrison and Li, 2008; Martin et al., 2010; Azetsu-Scott et al.,





2012). Where measured in the present study, $NO_2^-$ concentrations were < 0.36 µM and $NH_4^+$ concentrations were < 1 µM,

except for six samples from the Labrador Basin regime, with $NH_4^+$ concentrations up to 2.5 µM (Fig. S3-S7). Thus, with the exception of those few samples, the overall low $NO_2^-$ and $NH_4^+$ concentrations should have little impact on fPW estimates.

**Figure 5:** $\delta^{18}O_{NO3}$ **vs.** $\delta^{15}N_{NO3}$ **by station and $NO_3^-$ concentration. Arrows denote isotopic fractionation associated with $NO_3^-$ assimilation (slope = 1), and differences in water mass N cycling histories.**

**3.3 $NO_3^-$ isotope ratio variability**

Isotope ratios of $NO_3^-$ were measured at all 25 stations for $\delta^{15}N_{NO3}$ and all but the three stations from the HUD-2005-016 expedition for $\delta^{18}O_{NO3}$. Depth profiles are shown in Fig. 3g and 3h. Patterns of isotopic variability are presented separately for near-surface and sub-euphotic zone waters below.



### 3.3.1 Near-surface isotope ratios of $NO_3^-$

For waters above $z_P$, $\delta^{15}N_{NO3}$ increased from background values of around 5-6 ‰ to maxima of 12 ‰ toward the surface (Fig. 3g). The $\delta^{18}O_{NO3}$ similarly increased from < 2 ‰ to 11 ‰ (Fig. 3h). Fig. 5 shows a cross plot of $\delta^{18}O_{NO3}$ vs. $\delta^{15}N_{NO3}$. The dense cluster of data centered around $\delta^{15}N_{NO3}$ = 5 ‰ and $\delta^{18}O_{NO3}$ = 2 ‰ represents deep waters (> $z_p$) discussed below. For any given station, the paired isotope data for shallow water extend approximately along lines of 1:1. The increase in isotopic ratios coincides with a decrease in $NO_3^-$ concentrations (Fig. 5), and an increase in chlorophyll, as interpreted from fluorescence

profiles (Fig S1-S5). Together, these patterns are consistent with coupled (identical) fractionation of $^{15}N$ and $^{18}O$ during $NO_3^-$ assimilation (Granger et al., 2004; Sigman et al., 2005).

To further demonstrate the effect of $NO_3^-$ assimilation on isotopic ratios, $\delta^{15}N_{NO3}$ and $\delta^{18}O_{NO3}$ are plotted against the natural logarithm of $NO_3^-$ concentrations in Fig. 6. The "kinks" in the relationships in Fig. 6 represent the base of the $NO_3^-$ assimilation

zone. To the left of the kinks, both $\delta^{15}N_{NO3}$ and $\delta^{18}O_{NO3}$ increase with decreasing $NO_3^-$, again consistent with coupled fractionation of $^{15}N$ and $^{18}O$ during $NO_3^-$ assimilation. Moreover, assuming a mainly vertical supply of nutrients to the euphotic zone, the isotopic composition of the $NO_3^-$ used in assimilation may be approximated by the minima in $\delta^{15}N$ and $\delta^{18}O$ at the kinks (Fig. 6; Rafter and Sigman, 2016; Peters et al., 2018). In this respect, the $\delta^{15}N$ of the assimilated $NO_3^-$ increases from lowest values at the Labrador Basin stations to highest values at the Baffin Bay stations. The minima in $\delta^{18}O_{NO3}$ data show the

opposite trend, with the lowest values at the Baffin Bay stations and highest values in the Labrador Basin. Reasons for these differences are explored below.

### 3.3.2 Sub-surface isotope ratios of $NO_3^-$

Below $z_P$, $\delta^{15}N_{NO3}$ ranged from 4.1 - 6.5 ‰ (Fig. 3) and varied significantly by water mass (1-way ANOVA; Table 1). The Atlantic-derived water masses (IW, LSW, NEADW, DSOW) sampled in the Hudson Strait and Labrador Basin regimes had

the lowest mean $\delta^{15}N_{NO3}$ (4.8 ± 0.3‰). This value is identical to the $\delta^{15}N_{NO3}$ of North Atlantic intermediate-depth waters; it represents the basin-scale N isotopic mass balance between relatively $^{15}N$-depleted in Atlantic subtropical thermocline water and Mediterranean Overflow Water, and relatively $^{15}N$-enriched $NO_3^-$ in Antarctic Intermediate Water (Marconi et al., 2015). The Pacific-influenced HW, as well as BBW sampled in Baffin Bay and Davis Strait, displayed the highest mean $\delta^{15}N_{NO3}$ (> 6 ‰). The elevated $\delta^{15}N_{NO3}$ in HW reflects its predominant origin in the western Arctic. At the entrance to the western Arctic,

Pacific-origin $NO_3^-$ propagating onto the Bering Shelf has an already high $\delta^{15}N_{NO3}$ (6.3 ‰; Lehmann et al., 2005). As Pacific waters flow across the productive Bering and Chukchi shelves, $NO_3^-$ becomes further isotopically enriched due to benthic coupled nitrification-denitrification (CPND), which results in the removal of isotopically light $NH_4^+$ from the system and the efflux of heavy $NH_4^+$ into the overlying water column (Granger et al., 2011; Brown et al., 2015). Subsequent water column nitrification leads to the characteristically high $\delta^{15}N_{NO3}$ signature of the western Arctic upper halocline (~ 8.0 ‰, Brown et al.,

2015; Granger et al., 2018; Fripiat et al., 2018). The $\delta^{15}N_{NO3}$ signature in BBW, on the other hand, is consistent with *in situ*





remineralization in deep Baffin Bay, as indicated by high AOU and nutrient concentrations. The high $\delta^{15}N_{NO3}$ (> 7.0 ‰; Fig. 3g) indicates that the POM exported to the deep Baffin Bay is largely fueled by Pacific-derived nutrients in northern Baffin Bay (Lehmann et al., 2019), given that the N isotopic composition of newly nitrified $NO_3^-$ largely reflects the signature of its source substrate. The LShW sampled on the Labrador shelf exhibited intermediate and more variable $\delta^{15}N_{NO3}$ signatures (5.4

± 0.7 ‰), which, as with the corresponding N* data, was consistent with mixing of HW and IW across the Labrador Shelf.

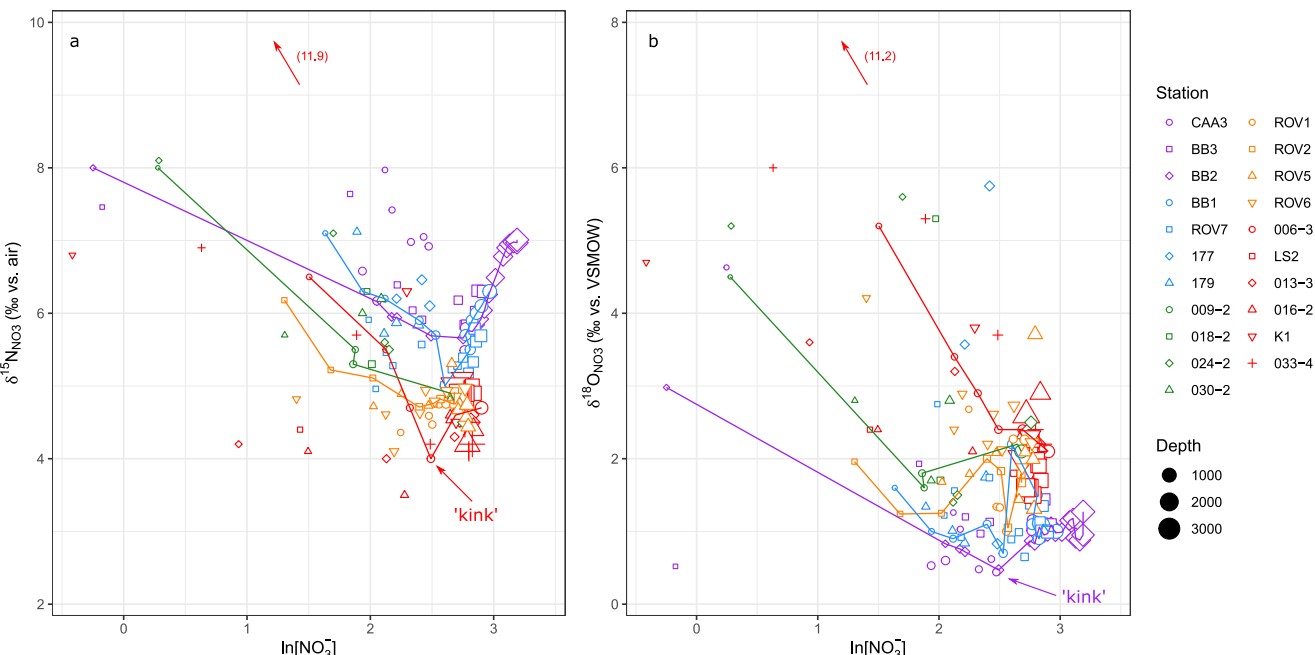

**Figure 6: (a) $\delta^{15}N_{NO3}$ and (b) $\delta^{18}O_{NO3}$ plotted against the natural logarithm of $NO_3^-$ concentrations. The 'kink' represents the base of the $NO_3^-$ assimilation zone. For clarity, only some of the station data are connected by lines.**


Values of $\delta^{18}O_{NO3}$ also varied significantly by water mass (1-way ANOVA) but in mirror image to $\delta^{15}N_{NO3}$ (Table 1). The LSW, NEADW, and DSOW exhibited higher $\delta^{18}O_{NO3}$ (> 1.8 ‰); HW and BBW had lower values (> 0.3 ‰). The low $\delta^{18}O_{NO3}$ in HW is within the range of values previously reported for the western Arctic upper halocline layer (Brown et al., 2015; Granger et al., 2018; Fripiat et al., 2018), where values close to 0 ‰ are indicative of the highly remineralized $NO_3^-$ pool, as

nitrification introduces a low $\delta^{18}O$ close to a value of ambient seawater (+1.1 ‰; Casciotti et al., 2008; Sigman et al., 2009; Buchwald et al., 2012). Low $\delta^{18}O_{NO3}$ values associated with BBW similarly reflect the high proportion of remineralized nutrients in deep Baffin Bay (Lehmann et al., 2019). The significantly higher $\delta^{18}O_{NO3}$ in the Labrador Sea subsurface layer reflects the remote signal of partial assimilation in the Southern Ocean, as well as a higher $\delta^{18}O$ of water oxygen atoms that are incorporated during remineralization in transit in the Atlantic versus the Arctic (Marconi et al., 2015; Granger et al., 2018).

The differential $NO_3^-$ isotope tagging of the various sub-euphotic water masses, which is a function of their different origin



and N-cycling history, holds great potential to trace the distribution of these water masses in the Northwest Atlantic and thus to assess the contribution from Pacific sources.

To explore the biogeochemical drivers of $\delta^{15}N_{NO3}$ and $\delta^{18}O_{NO3}$ in the different water masses further, a correlation matrix of physical and chemical variables was constructed (Fig. S8). The strongest covariates of $\delta^{15}N_{NO3}$ were fPW (r = 0.89) and N* (r = -0.86), followed by θ (r = -0.75), then variables associated with diatom and POM remineralization in BBW: $Si(OH)_4$ (r = 0.65), AOU (r = 0.64), and $PO_4^{3-}$ (0.61). Upon recalculating the correlation matrix *without* BBW (see below), the correlations with fPW (r = 0.91) and N* (r = -0.89) became even stronger, followed by salinity (r = -0.82), θ (r = -0.81) and $NO_3^-$ (r = -0.65). All five of these parameters exhibit multicollinearity, that is, waters with high fPW also have low N*, and are colder,
fresher, and have less preformed $NO_3^-$ than waters with low fPW. The same parameters were also correlated with $\delta^{18}O_{NO3}$ but opposite in sign.

    Figure 7a shows $\delta^{15}N_{NO3}$ plotted against N*. The regression line through the main group of data is highly significant ($p \ll$ 0.001) with an $r^2$ value of 0.78. Note that BBW data plot above and to the right of the rest of the data. We hypothesize that this
shift arises from remineralization of $PO_4^{3-}$ and $NO_3^-$, followed by loss of the $NO_3^-$ via sedimentary denitrification (Lehmann et al., 2019). A conceptual model of this two-step process is shown in the Fig. 7a inset. The source of preformed nutrients in the deep Baffin Bay is still debated (Tang et al., 2004), but, assuming a dominantly Atlantic source (Azetsu-Scott et al., 2012), the preformed nutrients would plot near the other Atlantic waters with N* values > 0 µM and $\delta^{15}N_{NO3} \sim$ +5 ‰ (e.g., Marconi et al., 2015; Granger et al., 2018; Fripiat et al., 2018). Under a simplifying assumption of near-Redfield stoichiometry, the N*
would remain unchanged during remineralization. (We note, however, that lower than Redfield N:P uptake has been documented in Baffin Bay (Harrison et al., 1982) which would shift the N* to lower values). The POM originates in the overlying $^{15}N_{NO3}$-enriched HW, which would generate remineralized $NO_3^-$ with relatively high $\delta^{15}N$ in BBW. Subsequent sedimentary denitrification would shift the N* to lower values, but would not further modify water column $\delta^{15}N_{NO3}$, as the $NO_3^-$ is completely reduced in the sediments (Brandes and Devol, 1997; Lehmann et al., 2004; 2007). This again highlights
that the processes affecting N* and $\delta^{15}N_{NO3}$ in BBW are separate and distinct from those influencing HW or the other water masses (Lehmann et al., 2019).

    Figure 7b shows $\delta^{15}N_{NO3}$ plotted against fPW, with the BBW data excluded because fPW cannot be calculated when $NO_3^-$ and $PO_4^{3-}$ ratios are not conserved. The regression line in Fig. 7b is also highly significant ($p \ll$ 0.001). with an $r^2$ of 0.80. The
intercept, corresponding to 100 % Atlantic water, is 4.8 ± 0.04 ‰. This value coincides exactly with previous estimates of the mean $\delta^{15}N_{NO3}$ (4.8 ‰) in North Atlantic intermediate and deep waters (Marconi et al., 2015). This is not surprising given that samples of 100 % Atlantic water are represented in the data distribution (Fig. 4). The regression also predicts the $\delta^{15}N_{NO3}$ for 100 % Pacific water at 8.3 ± 0.2 ‰. This value likewise matches with previous measurements of the Pacific halocline water measured in the eastern Beaufort Sea, downstream of the centers of CPND ($\delta^{15}N_{NO3}$ = 8.0 ± 0.1 ‰; Brown et al., 2015; Granger





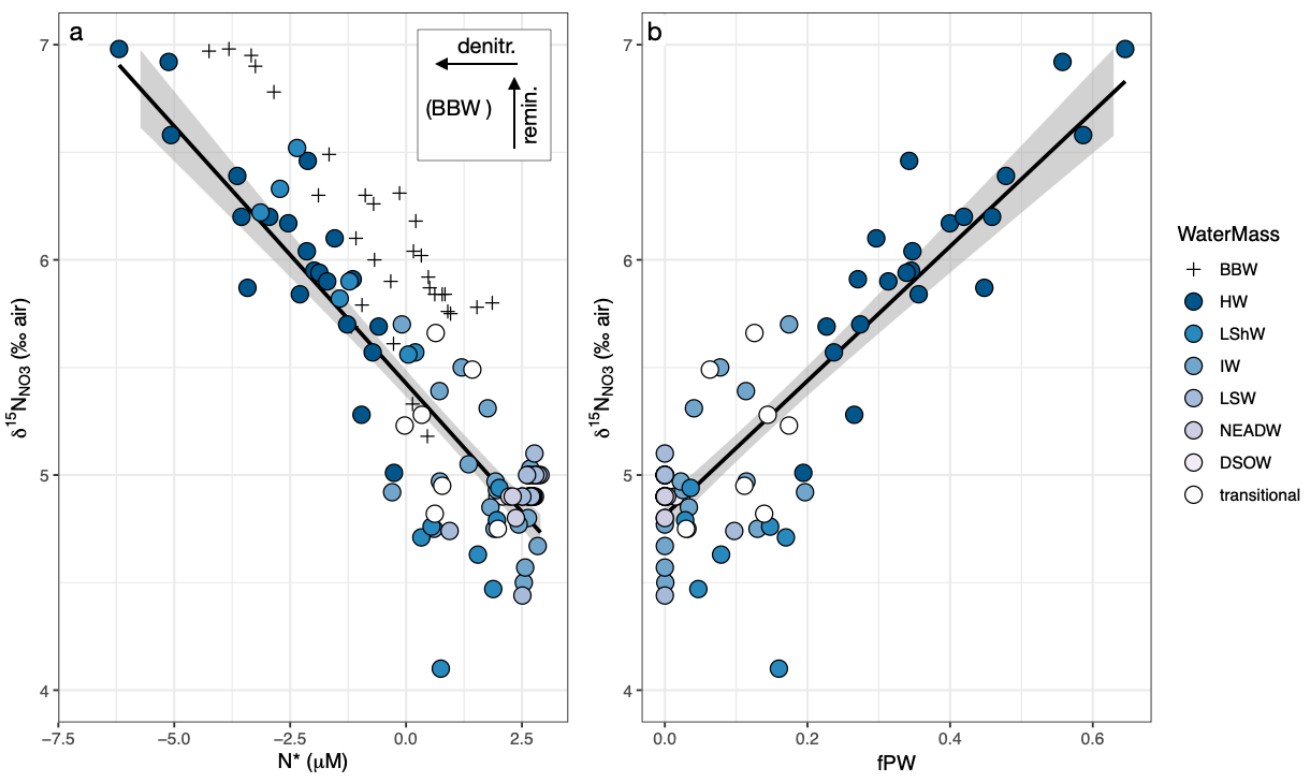

**Figure 7: (a) $\delta^{15}N_{NO3}$ vs. N\*. Regression line excludes BBW because these waters are affected by remineralization and denitrification as indicated by inset conceptual schematic. See text for explanation. (b) $\delta^{15}N_{NO3}$ vs. fPW, with BBW excluded because it is not possible to calculate fPW when $NO_3^-$ and $PO_4^{3-}$ ratios are not conserved. Data in both plots are for depths > $z_p$ to avoid the effects of $NO_3^-$ assimilation. Regression lines in both plots are bounded by 95 % confidence intervals.**

et al., 2018). Such accurate prediction of the Pacific endmember is remarkable, considering the degree of extrapolation beyond the limit (fPW ≤ 0.6) of sample data.

Relationships for $\delta^{18}O_{NO3}$ vs. N\* and $\delta^{18}O_{NO3}$ vs. fPW (not shown) were also highly significant (p << 0.001), but weaker, with $r^2$ values of 0.16 and 0.15, respectively. The $\delta^{18}O_{NO3}$ vs. fPW relationship predicts $\delta^{18}O_{NO3} = 2.0 \pm 0.1$ ‰ for 100 % Atlantic water and $\delta^{18}O_{NO3} = -0.1 \pm 0.5$ ‰ for 100 % Pacific water, close to previous direct measurements of Atlantic and Pacific water in their respective source regions (Marconi et al., 2015; Brown et al., 2015; Granger et al., 2018).

### 3.4 Preservation of $NO_3^-$ isotope signatures

One of the objectives of this paper is to assess the preservation of N isotopic signatures during transit of Pacific water from the Canadian Arctic Archipelago southward into the Northwest Atlantic. Accurate prediction of isotopic ratios in Atlantic and





Pacific source waters based on the relationships in Fig. 7b is perhaps the strongest indication that the signatures are well preserved. This preservation likely is facilitated by the oxic conditions in all water masses, as well as the extreme vertical density gradient, which isolates the HW from vertical mixing as it propagates through the Arctic (Tremblay et al., 2015). We also consider that $N_2$-fixation would act to increase N* while decreasing $\delta^{15}N_{NO3}$, and while there is some evidence of

diazotrophy in northern waters (Blais et al., 2012; Sipler et al., 2017; Harding et al., 2018) reported rates are small to negligible, and are therefore unlikely to impact water column $\delta^{15}N_{NO3}$ signatures, as suggested by the data in Fig. 7a.

### 3.5 Prediction of fraction Pacific water from $\delta^{15}N_{NO3}$

Another objective of this study is to evaluate the use of $\delta^{15}N_{NO3}$ as a water mass tracer. Based on the preceding results, sub-euphotic zone $\delta^{15}N_{NO3}$ may be considered as the product of two-endmember mixing of Atlantic and Pacific water (Fig. 7b).

Thus, by inversion of the linear regression in Fig. 7b, fPW may be derived. To achieve normality of model residuals it was necessary to remove data where fPW = 0, as well as three outliers identified in quantile-quantile plots. The resulting final model follows Eq. (7):

fPW = 0.23 ± 0.02 × $\delta^{15}N_{NO3}$ - 1.06 ± 0.08, ($r^2$ = 0.83, p << 0.001, n = 46).                    (7)

The 95 % confidence intervals on fPW predictions range from 0.02 to 0.1, which is roughly equal to the error associated with

estimating fPW from $NO_3^-$ vs. $PO_4^{3-}$ data (section 3.3; see also Jones et al., 2003). In other words, $\delta^{15}N_{NO3}$ may be used to estimate fPW about as well as nutrient data. Of course, it is less labour intensive to analyze and use N:P concentrations to estimate fPW, but there are scenarios in which a proxy $\delta^{15}N_{NO3}$ approach is complementary or even essential. For example, $\delta^{15}N_{NO3}$ of a water mass integrates the overall history of nutrient cycling, thereby smoothing out short-lived contingencies inherent to seawater chemistry data. N:P concentration data alone also cannot identify and/or distinguish between the various

processes (i.e., remineralization, nitrification, denitrification, diazotrophy) affecting nutrient concentrations and stoichiometric ratios. As a case in point, the two-step process of remineralization followed by sedimentary denitrification in BBW would not be obvious without isotopic data, and invites further investigations to help clarify their proportional effects on bottom water N:P ratios. We also suggest that isotope data can be used to screen samples that deviate from a two-end member mixing model for the calculation of fPW. Finally, the relationship in equation 7 provides, for the first time, a coherent framework for

interpreting $\delta^{15}N$ signatures incorporated into living and paleo-organic materials in the hydrographically complex Northwest Atlantic marine ecosystem.

### 3.6 From $\delta^{15}N_{NO3}$ to "baseline" $\delta^{15}N$

In isotope ecology and paleoceanography contexts, the term "baseline" $\delta^{15}N$ usually refers to the $\delta^{15}N$ of primary producer (phytoplankton) biomass. This baseline $\delta^{15}N$ signature is propagated to organisms higher up in the food web, overprinted by

trophic fractionation, which is often assumed to be about +3.4 ‰ per trophic level but in fact widely variable (Minagawa & Wada, 1984; Vander Zanden and Rasmussen, 2001). The baseline signature may also be altered by bacterial degradation during sinking and sedimentation of particulate organic material (Lehmann et al., 2002; Robinson et al., 2012). In either instance, it





is critical to know, or be able to approximate, baseline $\delta^{15}N$ in order to interpret the environmental significance of $\delta^{15}N$ variability recorded in organisms or sediments.


Phytoplankton fractionate against the heavier isotopes of N and O during growth on $NO_3^-$. Thus, under open-system conditions, the $\delta^{15}N$ of the phytoplankton will be lower than that of the $NO_3^-$. The isotopic fractionation varies from about 2 - 10 ‰, depending on phytoplankton species and growth conditions (Needoba et al., 2003). Under the semi-closed conditions of $NO_3^-$ drawdown in the ocean euphotic zone, the $\delta^{15}N$ of $NO_3^-$ and phytoplankton both increase, following Rayleigh fractionation

kinetics (Fig. 6). If the $NO_3^-$ is exhausted the $\delta^{15}N$ of phytoplankton will, by isotope mass balance, converge on that of the original, unassimilated $NO_3^-$. Hence, baseline $\delta^{15}N$ reflects the combined influences of $\delta^{15}N_{NO3}$ and the degree of $NO_3^-$ utilization (Altabet et al., 1999; Trull et al., 2008). It is difficult to distinguish between these influences unless $\delta^{15}N$ is measured on paired samples of phytoplankton and $NO_3^-$. This has not been done anywhere in our study region, apart from studies located in more inshore areas (Ostrom et al., 1997). Nevertheless, it is possible to make general inferences about nutrient drawdown

and its effect on baseline $\delta^{15}N$ from a consideration of regional nutrient-plankton bloom dynamics. In the Labrador Sea and Baffin Bay, light is the principal limiting factor to phytoplankton growth for most of the year; however, during the peak summer growth period, $NO_3^-$ becomes co-limiting or limiting as concentrations within the mixed layer are depleted (Harrison and Li, 2008). This applies even in the more light-limiting Arctic, where productivity is tightly coupled to $NO_3^-$ availability (Tremblay et al., 2006; Martin et al., 2010). Therefore, the $\delta^{15}N$ of the accumulated phytoplankton biomass should approach

that of the pre-assimilated $NO_3^-$, as identified by the "kink" in Fig. 6.  This was confirmed in a study of spring bloom dynamics in the North Water Polynya in northern Baffin Bay, where the $\delta^{15}N$ of phytoplankton converged on the $\delta^{15}N_{NO3}$ of Arctic halocline water (8.3 ‰) as the fraction of unassimilated $NO_3^-$ was drawn down to < 10 % of the pre-bloom concentrations (Tremblay et al., 2006). Considering that northern Baffin Bay is located at a latitude of maximum light limitation, we would predict that the patterns observed there also apply to Pacific-influenced waters of the more southerly Baffin Bay and continental

shelves of eastern Canada, except perhaps for inshore and upwelling regions, where $NO_3^-$ would be less limiting. For Atlantic-influenced waters, $NO_3^-$ is already relatively less limiting than P and Si (Fig. 3). Under these conditions, phytoplankton will be more likely to fractionate against $^{15}N$ (becoming "lighter"), thereby amplifying the existing differences in $\delta^{15}N_{NO3}$ between the Atlantic and Pacific derived water masses (Fig. 6). Additional studies are needed to determine the effective fractionation, if any, over seasonal and longer time scales.

**3.7 Implications for isotope ecology**

Results presented here may help to explain previously documented spatial variability in organism $\delta^{15}N$ in the Northwest Atlantic and Baffin Bay regions. For example, Sherwood and Rose (2005) examined bulk $\delta^{15}N$ in invertebrates and fish in waters off Newfoundland and Labrador. Organism $\delta^{15}N$ within each feeding guild was consistently higher, by up to 2.7 ‰, at coastal sites compared to shelf break sites. Part of this offset may be explained by the cross-shelf gradient in fPW, which

increases from < 0.02 at the shelf break to > 0.5 at the coast (Pepin et al., 2013; see also the fPW and $\delta^{15}N_{NO3}$ profiles for



stations 143, 147, and 154, Fig. S5), and corresponds to an increase of 2.3 ‰ in $\delta^{15}N_{NO3}$ based on Eq. (7). Similarly, in studies located off western Greenland, Hansen, J.H. et al. (2012) and Hedeholm et al. (2012) reported that $\delta^{15}N$ in primary consumers increased by 2 - 3 ‰ over a latitudinal gradient from 60 - 72°N. Subsurface fPW off southern Greenland is essentially zero (Sutherland et al., 2009; Azetsu-Scott et al., 2012), and increases northward as IW and West Greenland Shelf Water mixes

with HW, reaching values > 0.4 based on sparse nearby data (e.g., Station ROV7 profile, Fig. S4; Hansen et al., 2012). The corresponding increase in $\delta^{15}N_{NO3}$ is > 2 ‰. This suggests that, in the examples above, the spatial variability in organism $\delta^{15}N$ may be attributed largely to the differential water mass partitioning, rather than spatial variations in the degree of $NO_3^-$ utilization directly at the respective study sites. Finally, Sherwood et al. (2008) examined the bulk $\delta^{15}N$ in the tissues of deep-sea corals collected along the continental slope from Hudson Strait (62°N) to the southwest Grand Banks of Newfoundland

(43°N). They found *no* overall change in $\delta^{15}N$ with respect to latitude, but this is consistent with the minimal latitudinal change in fPW (< 0.1) along the path of the slope component of the Labrador Current (Jones et al., 2003). Overall, these examples reiterate the fundamental importance of accounting for variability in baseline $\delta^{15}N$ in isotope ecology studies (e.g., de la Vega et al., 2021). It is not always feasible to measure $\delta^{15}N$ in $NO_3^-$ or primary producers directly, thus we suggest that baseline $\delta^{15}N$ for Canadian Arctic and Northwest Atlantic region may be approximated, to a first degree, from nutrient concentrations and

either of the N* or fPW relationships presented in Fig. 7.

### 3.8 Implications for paleoceanography

Our results also have important implications for regional paleoceanographic interpretations of $\delta^{15}N$ as recorded in sedimentary organic matter and in long-lived biological archives. With respect to sediments, $\delta^{15}N$ is confounded by site-to-site differences in sedimentation rates and diagenetic effects (Robinson et al., 2012). Nevertheless, known spatial patterns track the expected

distribution of fPW, with lower values of 4 - 6 ‰ in the central Labrador Sea and Southwest Greenland margin, and higher values 6 - 9 ‰ on the Labrador shelf and northern Baffin Bay (Muzuka & Hillaire-Marcel, 2000; Cormier et al., 2016; Kienast et al., 2020; Limoges et al., 2020). Thus, by extension, downcore trends in $\delta^{15}N$ should reflect advection-related temporal changes in fPW. Based on arguments in section 3.6, this advection influence is likely to exceed the influence of surface water $NO_3^-$ utilization, particularly where $NO_3^-$ is limiting. This may help to explain why downcore variations in $\delta^{15}N$ are positively

correlated with other, biomarker and micropaleontological proxies of Arctic throughflow to Baffin Bay (Cormier et al. 2016; Limoges et al. 2020), confirming the potential of sedimentary $\delta^{15}N$ to quantitatively reconstruct changes in fPW in the past, at least in areas where local changes in nutrient utilization did not play a greater role. This also applies to records of $\delta^{15}N$ recorded in biological archives such as deep-sea corals, which have been shown to track changes in the southward advection of the Labrador Current over the 20th century (Sherwood et al., 2011). We note that, as the N-cycling regimes in the source region

and/or in the North Atlantic may have shifted in the past, long term changes in downcore or archival $\delta^{15}N$ may also be influenced by variability in endmember $\delta^{15}N_{NO3}$ signatures (i.e. $NO_3^-$ "inventory-altering" effects; Galbraith et al. 2013), particularly for the Pacific water endmember which is sensitive to primary productivity via sedimentary CPND on the western





Arctic shelves. Thus, long term variability in $\delta^{15}N$ should be carefully interpreted in the context of all three influences - nutrient utilization, advection, and changes to endmember $\delta^{15}N_{NO3}$ signatures - together with other lines of paleoceanographic evidence.

**4. Conclusions**

The flow of Pacific water through the Canadian Arctic Archipelago and into the Northwest Atlantic plays a key role in global thermohaline circulation and biogeochemical cycling. The isotopic composition of $NO_3^-$ presents a new way to track this influence, expanding on the existing N:P stoichiometry approach. Isotopically distinct Pacific water ($\delta^{15}N_{NO3}$ = 8.3 ‰; $\delta^{18}O_{NO3}$ = 0 ‰) travels as a subsurface halocline layer through the Canadian Arctic Archipelago, and onward to Labrador Shelf, with

little apparent alteration other than mixing with Atlantic water ($\delta^{15}N_{NO3}$ = 4.8 ‰; $\delta^{18}O_{NO3}$ = 2.0 ‰). The resulting two-endmember mixing of Pacific and Atlantic water is described by a new empirical relationship that may be used to estimate the fraction of Pacific water from $\delta^{15}N_{NO3}$. The deep waters of Baffin Bay are distinctly different, with nutrient inventories showing an imprint of both *in situ* remineralization and sedimentary denitrification. These deep waters are isolated below 500 m and therefore do not influence baseline $\delta^{15}N$ incorporated into primary producer biomass. Rather, baseline $\delta^{15}N$ throughout the

Labrador-Baffin region should primarily reflect the fraction Pacific water, particularly where $NO_3^-$ is the limiting nutrient. Overall, these results provide a new framework for interpreting spatial and temporal patterns of $\delta^{15}N$ in isotope ecology and paleoceanography contexts. In particular they highlight the potential of $\delta^{15}N$ recorded in sedimentary and organic paleo-archives to quantitatively reconstruct changes in fPW in the past.

**Supplement link**

Supplementary information accompanies this manuscript.

**Author contributions**

OS and SD conceptualized the research with input from all authors. OS, SD and MK collected the samples. SD and NL carried out nitrate isotope analyses. OS, SD, NL, CB and MK analyzed the data. OS, SD and NL prepared the manuscript with contributions from all authors.

**Competing interests**

The authors declare that they have no conflicts of interest



**Acknowledgements**

We thank the captain, crew and science staff of the R/V *Maria S. Merian* MSM45 and CCGS *Amundsen* AMD2016 expeditions. We also thank ArcticNet (a Canadian Network of Centres of Excellence), and Amundsen Science for their in-
kind contributions to expedition logistics and scientific equipment. McKenzie Mandich (Dalhousie University) analyzed nutrient concentrations for the MSM45 samples. Jean-Éric Tremblay facilitated sample collection and nutrient analysis during the AMD2016 expedition. Thomas Kuhn (University of Basel) analyzed nitrate isotopes for the AMD2006 samples. Claude Hillaire-Marcel provided input on earlier drafts of the manuscript. Funding for this project was provided by the Canadian Foundation for Innovation, and the Natural Sciences and Engineering Research Council (NSERC) of Canada through a Climate
Change and Atmospheric Research (CCAR) grant to Paul Myers et al. (2013-2018), a ship time (STAC) grant to EE et al., a Strategic Projects grant to MK, OS, et al., and Discovery Grants to OS, CB, EE, MK and Claude Hillaire-Marcel.

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
