# Peer review of "Stable isotope ratios in seawater nitrate reflect the influence of Pacific water along the Northwest Atlantic margin"

_Biogeosciences, 2021_

## Referee Comment (RC2)

**General comments:**

This is a well-written and relatively concise manuscript that uses $\delta^{15}N$ of nitrate data to trace the distribution of Pacific versus Atlantic waters in the Northwest Atlantic. This type of analysis is not new, the manuscript by Granger et al. (2018) previously laid the groundwork. However, while the focus of these two papers is similar, the current manuscript presents new relationships that estimate the fraction of Pacific water (based on both N* and the $\delta^{15}N$ of nitrate) using a more extensive dataset and discuss some possible applications (food-web studies, paleoceanographic reconstructions, etc…). I have some minor comments to improve the manuscript. First, it would be best to separate the results from the discussion to improve clarity and focus, if at all possible. Most of the text before section 3.4 could be moved to a Results section, as these sections are mostly descriptive, the remainder could be re-organized into a proper discussion section. Second, some analytical detail or background information in the discussion needs to be added (see specific comments below). Finally, I wonder if the $\delta^{18}O$ data could be explored in more detail. These data are shown in Figure 3, but poorly discussed in the manuscript.

**Specific comments:**

**Materials and methods:**

Lines 96-107: Is using different types of filters (0.45 $\mu m$ versus 0.22 $\mu m$) affect nutrient concentrations? Was this tested?

Lines 121-122: Was USGS 32 used to correct $\delta^{15}N$ data? Since its $\delta^{15}N$ is much different from the $\delta^{15}N$ of the samples, I assume this would be problematic.

Lines 123-125: Why was $NO_2^-$ not removed? Even small $NO_2^-$ concentrations can affect the $\delta^{15}N$ and $\delta^{18}O$ of $NO_3^-$, especially at low $NO_3^-$ concentrations. What was the lowest $NO_3^-$ concentration for samples analyzed for isotopic composition? Since nitrate concentrations are generally high below the mixed layer, and for most water masses discussed in this manuscript, I don't think this is a major concern.

Another concern is that the "denitrifier" method is sensitive (to some extent) to the $\delta^{18}O$ value of the sample water because of O exchange with water during the conversion of $NO_3^-$ to $N_2O$. This is an issue if the $\delta^{18}O$ of the samples and standards are drastically different or in polar regions where the $\delta^{18}O$ of water is greatly variable due to mixing between freshwater from rivers and glaciers (with a low $\delta^{18}O$ around -20 ‰) and seawater ($\delta^{18}O$ of about 0‰). Was this taken into account while analyzing their samples? See Kobayashi et al. (2021) for more detail. Finally, what was their blank size (i.e., for the bacteria method)?

Lines 144-145: The authors should make a clearer distinction between regenerated (calculated using Redfield $P:O_2$ ratio and AOU) versus preformed $PO_4^{3-}$ here. Preformed nutrients are those

that were present in solution when the parcel of water sank from the surface and are characteristic of different water masses:

pre-formed $[PO_4^{3-}]$ = measured $[PO_4^{3-}]$ + regenerated $[PO_4^{3-}]$

Lines 160-161: Explain what cause that kink in the $NO_3^-$ vs $PO_4^{3-}$ relationship at low nutrient concentrations (i.e., nitrate assimilation).

**Results and Discussion:**

Lines 367-371: I am curious about the isotope effect for nitrate assimilation derived from these relationships and how it compares to previous field studies (e.g., Altabet et al. (2001)?

Lines 415-416: Why isn't the correlation showed for $\delta^{18}O-NO_3^-$? Could $\delta^{18}O$ of $NO_3^-$ also be used as a complementary tool to trace these different water masses? This aspect should be better discussed.

Lines 427-429: The authors should explain why no change in water-column $\delta^{15}N-NO_3^-$ is expected during sedimentary denitrification (i.e., discuss the suppressed net "community" isotope effect for sedimentary denitrification due to diffusion limitation and complete $NO_3^-$ consumption in the sediments).

Line 468: Could a similar equation be derived for $\delta^{18}O$ of $NO_3^-$ as well? However, $\delta^{18}O$ of $NO_3^-$ would not be useful for food-web or paleoceanographic studies as the O atom is not conserved during N incorporation into organic material.

Lines 471-478: This argument needs to be discussed better since as for N*, it is not possible to disentangle different co-occurring processes (nitrification, denitrification, $N_2$ fixation) using $\delta^{15}N$ of $NO_3^-$ data solely. These co-occurring processes were disentangled in the BBW because of the additional insights from $\delta^{18}O$ of $NO_3^-$ data.

Line 483: This section title is vague. I would rename it "Using our $\delta^{15}N-NO_3^-$ relationship to establish a baseline $\delta^{15}N$ for food-web and paleoceanographic studies." This section could also be merged with sections 3.7 and 3.8.

Line 565: Change to "fraction of Pacific water"

**Table 1.** Add number of samples analyzed for each water masses (n).
Indicating a range of depths for each water masses would be better than showing the average (given the large standard deviation).

**Figure 3.** Are $\delta^{18}O$ of $NO_3^-$ values shown at about 200 m depth (177) and 500 m depth (ROV5) outliers? It is unclear why there is no corresponding increase in the $\delta^{15}N$ of $NO_3^-$ at these stations/depths. Were these samples measured in duplicate?

**Figure 5.** I think it would make sense to separate the symbols based on depths for this figure (e.g., as in Figure 2: surface waters impacted by nitrate assimilation (open symbols) versus deeper waters (filled symbols)).

**Figure 7.** The $R^2$ as well as p-values should be added.

**Additional references:**

Altabet, M. A. (2001). Nitrogen isotopic evidence for micronutrient control of fractional NO3– utilization in the equatorial Pacific. *Limnology and oceanography*, *46*(2), 368-380.

Kobayashi, K., Fukushima, K., Onishi, Y., Nishina, K., Makabe, A., Yano, M., ... & Okabe, S. (2021). Influence of δ18O of water on measurements of δ18O of nitrite and nitrate. *Rapid Communications in Mass Spectrometry*, *35*(2), e8979.

---

## Author Comment (AC2)

**Response to Reviewer 2**

***R2.1 General comments:***

*This is a well-written and relatively concise manuscript that uses $\delta^{15}N$ of nitrate data to trace the distribution of Pacific versus Atlantic waters in the Northwest Atlantic. This type of analysis is not new, the manuscript by Granger et al. (2018) previously laid the groundwork. However, while the focus of these two papers is similar, the current manuscript presents new relationships that estimate the fraction of Pacific water (based on both N\* and the $\delta^{15}N$ of nitrate) using a more extensive dataset and discuss some possible applications (food-web studies, paleoceanographic reconstructions, etc…).*

- We thank the reviewer for taking the time to read the manuscript and for offering their comments and constructive criticisms.

***R2.2*** *I have some minor comments to improve the manuscript. First, it would be best to separate the results from the discussion to improve clarity and focus, if at all possible. Most of the text before section 3.4 could be moved to a Results section, as these sections are mostly descriptive, the remainder could be re-organized into a proper discussion section.*

- We generally agree with this suggestion. Sections 3.1 to 3.3 contain mostly descriptive results and can be re-titled as "Results" (with existing subheadings), with some moving of the more inferential statements to a new Discussion section.  Sections 3.4 to 3.8 will be re-titled as "Discussion" with some minor re-organization.

***R2.3*** *Second, some analytical detail or background information in the discussion needs to be added (see specific comments below).*

- Please see our responses to points R2.5 to R2.9 below. These details will be added to section 2.2 of the Materials and Methods.

***R2.4*** *Finally, I wonder if the $\delta^{18}O$ data could be explored in more detail. These data are shown in Figure 3, but poorly discussed in the manuscript.*

- Respectfully, a deeper exploration of the $\delta^{18}O_{NO3}$ data would require a more detailed discussion of the factors that modulate $\delta^{18}O_{NO3}$ variability, which would go beyond the scope of the ms. The primary focus is on $\delta^{15}N_{NO3}$ variability because it is preserved in organic materials and therefore important in isotope ecology and paleoceanography contexts. The $\delta^{18}O_{NO3}$ data help to *support* the interpretations of $\delta^{15}N_{NO3}$ variability (Figures 5 and 6 and associated discussion) and provide additional insights as for example the interpretations of sedimentary denitrification in BBW (Lehmann et al. 2019), but the $\delta^{18}O_{NO3}$ in isolation are not a diagnostic tracer for Pacific water.  We propose to rephrase the final paragraph of the introduction to clarify that the focus is on $\delta^{15}N_{NO3.}$

***Specific comments:***

***R2.5 Materials and methods:***

*Lines 96-107: Is using different types of filters (0.45 $\mu$m versus 0.22 $\mu$m) affect nutrient concentrations? Was this tested?*

- The use of different filter sizes was an unintended consequence of obtaining samples opportunistically during different expeditions. However, most organisms that could impact nutrient concentrations between the point of collection and subsequent freezing and analysis are larger than 0.45 $\mu$m.  We did not perform any specific tests, but there is no indication that any data were impacted by the small difference in filter pore size. We also note that 0.45 $\mu$m filters are used regularly for nutrient measurements.

***R2.6*** *Lines 121-122: Was USGS 32 used to correct $\delta^{15}N$ data? Since its $\delta^{15}N$ is much different from the $\delta^{15}N$ of the - samples, I assume this would be problematic.*

- The reviewer is correct in noting that the USGS 32 standard lies beyond the range of sample $\delta^{15}N_{NO3}$ values. It is used in a 3-point calibration as part of routine operating procedures in the Dalhousie lab. Omitting USGS 32 from the calibration curve had negligible impact (<0.2‰ ) on sample $\delta^{15}N_{NO3}$ values.

**R2.7** *Lines 123-125: Why was $NO_2^-$ not removed? Even small $NO_2^-$ concentrations can affect the $\delta^{15}N$ and $\delta^{18}O$ of $NO_3^-$, especially at low $NO_3^-$ concentrations. What was the lowest $NO_3^-$ concentration for samples analyzed for isotopic composition? Since nitrate concentrations are generally high below the mixed layer, and for most water masses discussed in this manuscript, I don't think this is a major concern.*

- The reviewer is correct in their last point about higher concentrations below the mixed layer in the water masses of concern. The $NO_2^-$ concentrations in this zone were always less than 0.36 μM, as shown in Figures S3 – S7, which is too low relative to the much higher $NO_3^-$ concentrations (< 6 μM) to affect the isotopic compositions. We report some isotope data from mixed layer waters with $NO_3^-$ concentrations as low as 0.7 μM, but since the focus is on waters below the mixed layer, this is not a major concern. We will clarify this point in the revision. We will also clarify that "$NO_3$" actually refers to the sum of $NO_3^-$ and $NO_2^-$ throughout the manuscript.

**R2.8** *Another concern is that the "denitrifier" method is sensitive (to some extent) to the $\delta^{18}O$ value of the sample water because of O exchange with water during the conversion of NO3- to N2O. This is an issue if the $\delta^{18}O$ of the samples and standards are drastically different or in polar regions where the $\delta^{18}O$ of water is greatly variable due to mixing between freshwater from rivers and glaciers (with a low $\delta^{18}O$ around -20 ‰) and seawater ($\delta^{18}O$ of about 0‰). Was this taken into account while analyzing their samples? See Kobayashi et al. (2021) for more detail.*

- We did not apply a correction for the $\delta^{18}O$ of seawater. However, the $\delta^{18}O$ in the sample waters range only from -2.2 ‰ to 0.2 ‰ (Lehmann et al. 2019) which would have a minor impact on the $\delta^{18}O_{NO3}$ results. We will note this in the revision.

**R2.9** *Finally, what was their blank size (i.e., for the bacteria method)?*

- The blank size constituted <2 % for the routine 20 nmol target analyses, and <5 % for the low concentration 5nmol target surface water analyses. We will state this in section 2.2.

**R2.10** *Lines 144-145: The authors should make a clearer distinction between regenerated (calculated using Redfield $P:O_2$ ratio and AOU) versus preformed $PO_4^{3-}$ here. Preformed nutrients are those that were present in solution when the parcel of water sank from the surface and are characteristic of different water masses:*
*pre-formed $[PO_4^{3-}]$ = measured $[PO_4^{3-}]$ + regenerated [PO43-]*

- Agreed. We will articulate a clearer distinction between preformed and regenerated $PO_4^{3-}$ in the revision.

**R2.11** *Lines 160-161: Explain what cause that kink in the $NO_3^-$ vs $PO_4^{3-}$ relationship at low nutrient concentrations (i.e., nitrate assimilation).*

- We will expand on the sentence at line 160 to explain the kink in the $NO_3^-$ vs $PO_4^{3-}$ relationship for Pacific water in the Canada Basin.

**Results and Discussion:**
**R2.12** *Lines 367-371: I am curious about the isotope effect for nitrate assimilation derived from these relationships and how it compares to previous field studies (e.g., Altabet et al. (2001)?*

- We calculated the apparent fractionation for $NO_3^-$ assimilation for the few stations with a sufficient number of shallow water $\delta^{15}N_{NO3}$ data, based on the slopes of $\delta^{15}N_{NO3}$ versus the negative natural logarithm of the fraction sub-euphotic zone $NO_3^-$ concentrations. The fractionation factors were smaller (0.8 to 3.9 ‰) than the generally assumed 5‰ fractionation for assimilation, but consistent with observations in Granger et al. (2010). However, we choose not to report these results, as they do not address the objectives of the manuscript.

**R2.13** *Lines 415-416: Why isn't the correlation showed for $\delta^{18}O$-$NO_3^-$? Could $\delta^{18}O$ of $NO_3$ also be used as a complementary tool to trace these different water masses? This aspect should be better discussed.*

- With respect to the first point, we will add the correlation matrices (as in figure S8) for $\delta^{18}O_{NO3}$ to the supplementary figures. With regard to discussion of the $\delta^{18}O_{NO3}$ data, please see our response to point R2.4 above.

**R2.14** *Lines 427-429: The authors should explain why no change in water-column $\delta^{15}$N-NO$_3^-$ is expected during sedimentary denitrification (i.e., discuss the suppressed net "community" isotope effect for sedimentary denitrification due to diffusion limitation and complete NO3-consumption in the sediments).*

- We will add a sentence to expand on our existing explanation for why sedimentary denitrification does not impact water column $\delta^{15}$N$_{NO3}$.

**R2.15** *Line 468: Could a similar equation be derived for $\delta^{18}$O of NO$_3^-$ as well? However, $\delta^{18}$O of NO$_3^-$ would not be useful for food-web or paleoceanographic studies as the O atom is not conserved during N incorporation into organic material.*

- The reviewer is correct in their assessment that $\delta^{18}$O$_{NO3}$ has little utility with respect to foodweb or paleoceanographic applications, in the sense that oxygen is not conserved during incorporation into biological materials. This is our rationale for not showing the relationship between $\delta^{18}$O$_{NO3}$ and fPW. We will add a sentence to make this clearer.

**R2.16** *Lines 471-478: This argument needs to be discussed better since as for N\*, it is not possible to disentangle different co-occurring processes (nitrification, denitrification, N$_2$ fixation) using $\delta^{15}$N of NO$_3^-$ data solely. These co-occurring processes were disentangled in the BBW because of the additional insights from $\delta^{18}$O of NO$_3^-$ data.*

- We propose to expand this section, tightening up our argument for the utility of $\delta^{15}$N$_{NO3}$ as a proxy for fPW.

**R2.17** *Line 483: This section title is vague. I would rename it "Using our $\delta^{15}$N-NO$_3^-$ relationship to establish a baseline $\delta^{15}$N for food-web and paleoceanographic studies." This section could also be merged with sections 3.7 and 3.8.*

- We will retitle section 3.6 accordingly. But we prefer to maintain the current structure of sections 3.6, 3.7 and 3.8.

**R2.18** *Line 565: Change to "fraction of Pacific water"*

- We agree that writing "fraction of Pacific water" instead of the abbreviated "fPW" in the conclusions will assist the reader. We will make this change.

**R2.19** *Table 1. Add number of samples analyzed for each water masses (n).*
*Indicating a range of depths for each water masses would be better than showing the average (given the large standard deviation).*

- We will incorporate both of these suggestions.

**R2.20** *Figure 3. Are $\delta^{18}$O of NO$_3^-$ values shown at about 200 m depth (177) and 500 m depth (ROV5) outliers? It is unclear why there is no corresponding increase in the $\delta^{15}$N of NO3- at these stations/depths. Were these samples measured in duplicate?*

- These are most likely analytical outliers, especially given the lack of a corresponding increase in $\delta^{15}$N$_{NO3}$. Unfortunately, neither of these samples was measured in duplicate. In the interests of data transparency, we choose to show these data. However, we will remove the dashed lines connecting these points to the rest of the profile data, which should help to improve figure clarity. We will explain this in the figure caption.

**R2.21** *Figure 5. I think it would make sense to separate the symbols based on depths for this figure (e.g., as in Figure 2: surface waters impacted by nitrate assimilation (open symbols) versus deeper waters (filled symbols)).*

- Respectfully, we disagree with this suggestion to reformat the symbols in Fig. 5. However, to improve clarity we propose to rescale the symbol sizes representing NO$_3^-$ concentrations. This will make it easier for the reader to infer which samples were impacted by NO$_3^-$ utilization.

**2.22** *Figure 7. The R2 as well as p-values should be added.*

- We will add $r^2$ and p-values to the plot, as requested.

---

## Author Response (AR1)

***Response to Reviewer 1:***
***R1.1*** *This study presents an extensive dataset focusing on N-dynamics along the NW Atlantic margin where water masses play an important role in nutrient distribution. The topic fits with the objectives of Biogeosciences even though there is the minimum amount of biology mentioned. Nitrogen and related nutrient dynamics are crucial elements to improve our understanding of biogeochemistry in the marine realm. Therefore, I think this study is an important input to our current knowledge on N-cycling.*

***R1.2*** *The manuscript is well-written, figures and tables are structured nicely and representative enough. The dataset is extensive and the structure chosen here for results & discussion complicates the reading a little bit. However, I am aware that such extensive information is difficult to present. Accordingly, I have few suggestions to improve the MS for the different target audiences (e.g., ecologists, paleoceanographers) and to make it a bit more reader-friendly.*
- We thank the reviewer for taking the time to read the manuscript and for offering their constructive criticisms.

***R1.3*** *Appendix with all the abbreviations used in the MS. Table 1 is really helpful, but if it fits with the journal regulations a list of all the abbreviations used would be nice.*
- We agree that the many abbreviations used throughout the ms can be cumbersome. Most of the abbreviations are used for referencing the seven different water masses, and are listed in Table 1. We will defer to the editor's recommendation as to whether an additional table of abbreviations would be appropriate.

***R1.4*** *Additional figure showing water masses in-depth with characteristics; e.g., NE-SW transect along the margin vs water depth showing $\delta N_{NO_3}$ (or other parameters to visualize the water masses in-depth and latitude). I am aware that Figure 3 aims and shows that, but I think such a transect would make it easier to visualize the different water mass dynamics and geographic distribution of stations for such a region. Station names could be also be shown on this transect figure.*
- We agree that section plots, in general, help to illustrate water mass distributions as a function of depth and distance. However, with the great geographic expanse and complex circulation in our study region, it would be difficult to portray the water masses in one section. The oceanography is best shown as a series of meridional transects, as for example in Fratantoni and Pickart (2007) Jones et al. (2003) and Tang et al. (2004). Moreover, the stations represented in our dataset were sampled opportunistically during four different expeditions and thus are not appropriately aligned along any particular transect. As a result, a section plot through any subset of our stations, whether oriented N-S or E-W, would produce an overly artificial representation of $\delta^{15}N_{NO3}$ variability. This would be particularly problematic in plotting a section along the margin, as the stations were located variously on- and off-shelf, which represent distinctly different oceanographic regimes. This is why we opted to show only the depth profiles (Fig. 3). The colour scheme with purple and blue for the more northerly/colder regions and orange and red for the more southerly/warmer regions is intended to help facilitate the visual distinction of depth profiles by region.

***R1.5*** *Ignore the use of sentences like "Figure XX shows this" e.g., lines 324 and 418. A reference to figures within the text should be sufficient. Accordingly, figure captions can be more informative and descriptive.*
- We will revise the phrasing of in-text figure references and improve the clarity of the figure captions, as suggested.

***R1.6*** *Do authors plan to store the dataset on a public platform? I highly encourage this.*
- Yes. The data were uploaded to Dryad at the time of submission, and a link is provided in the manuscript "Assets" page via: https://bg.copernicus.org/preprints/bg-2021-45/

***R1.7*** *Abstract: Line 20: change N/P to N:P*
- We thank the reviewer for catching this inconsistency. We will correct this and all other inconsistencies throughout the manuscript.

***R1.8*** *Introduction:*
*I recommend changing the structure of the introduction. If the target audience is ecologists and paleoceanographers, I would start with a short introduction of the use of 15N in these fields and then focus on the region; why here? And later on, give this regional information that is now at the start of the section.*

- We propose to revise the introductory paragraph and merge it with paragraph 2. We will more briefly introduce the fact that Pacific water constitutes an important fraction of the slope and shelf waters of the NW Atlantic, but move the more specific details of the circulation pathways to section 3.1.

***R1.9*** *The current structure of the introduction; starting right away with water masses in the study area, also requires a reference to Figure 1. For someone interested in N, particularly in such a dynamic system, I find the current structure of the introduction is distracting.*

- Agreed. A reference to Fig. 1 will be added to the introductory paragraph.

***R1.10*** *The last paragraph of the section (starting from line 74): This part needs more information on the overall objectives of the MS including ecologic perspective as well as mentioned in the beginnings of sections 3.4 and 3.5 for instance.*

- We will elaborate and expand on the goal and objectives with particular reference to the topics discussed in sections 3.4 through 3.8.

***R1.11*** *Results & Discussion:*
*Full of information and well-designed in terms of structure. As I mentioned above, the description of figures shouldn't be given in the text though. If the figure captions are improved then such sentences (Line 324-326) could be removed from this part and the overall text can be simplified.*

- We thank the reviewer for the positive appraisal of the structure of the Results and Discussion.
- We will revise the phrasing of in-text figure references and improve the clarity of the figure captions as suggested.

***R1.12*** Does "near-surface" mentioned in subsections always consider the same water depths? E.g., in section 3.2.1 I am missing the information on Zp.

- We agree that the terms "near surface" and "sub-surface" are unclear. We will change the title of section 3.2.1 to "Nutrient concentrations in the biologically productive zone" and we will reiterate that this zone refers to depths < Zp, as defined in section 2.3. Similarly, we will change the title of section 3.2.2 to "Nutrient concentration below the biologically productive zone" and clarify that this depth encompasses depths > Zp.

***R1.13*** Why are $\delta^{18}O_{NO3}$ *results not shown at all? I think it is worth mentioning them in the supplementary material.*

- The $\delta^{18}O_{NO3}$ data are presented in Table 1, and in Figures 3, 5, and 6, and discussed in sections 3.3.1 and 3.3.2. But please see our response to Reviewer 2 regarding our discussion of the $\delta^{18}O_{NO3}$ data.

**Response to Reviewer 2**

**R2.1 General comments:**

*This is a well-written and relatively concise manuscript that uses $\delta^{15}N$ of nitrate data to trace the distribution of Pacific versus Atlantic waters in the Northwest Atlantic. This type of analysis is not new, the manuscript by Granger et al. (2018) previously laid the groundwork. However, while the focus of these two papers is similar, the current manuscript presents new relationships that estimate the fraction of Pacific water (based on both N\* and the $\delta^{15}N$ of nitrate) using a more extensive dataset and discuss some possible applications (food-web studies, paleoceanographic reconstructions, etc...).*

- We thank the reviewer for taking the time to read the manuscript and for offering their comments and constructive criticisms.

**R2.2** *I have some minor comments to improve the manuscript. First, it would be best to separate the results from the discussion to improve clarity and focus, if at all possible. Most of the text before section 3.4 could be moved to a Results section, as these sections are mostly descriptive, the remainder could be re-organized into a proper discussion section.*

- We have separated the "Results" and "Discussion" sections and renumbered the sub-headings accordingly.

**R2.3** *Second, some analytical detail or background information in the discussion needs to be added (see specific comments below).*

- Please see our responses to points R2.5 to R2.9 below.

**R2.4** *Finally, I wonder if the $\delta^{18}O$ data could be explored in more detail. These data are shown in Figure 3, but poorly discussed in the manuscript.*

- Respectfully, a deeper exploration of the $\delta^{18}O_{NO3}$ data would require a more detailed discussion of the factors that modulate $\delta^{18}O_{NO3}$ variability, which would go beyond the scope of the ms. The primary focus is on $\delta^{15}N_{NO3}$ variability because it is preserved in organic materials and therefore important in isotope ecology and paleoceanography contexts. The $\delta^{18}O_{NO3}$ data help to *support* the interpretations of $\delta^{15}N_{NO3}$ variability (Figures 5 and 6 and associated discussion) and provide additional insights as for example the interpretations of sedimentary denitrification in BBW (Lehmann et al. 2019), but the $\delta^{18}O_{NO3}$ in isolation are not a diagnostic tracer for Pacific water. To that end, we have tightened up the last paragraph of the introduction to better reflect that the primary focus of the ms is on $\delta^{15}N_{NO3}$. We have also expanded on the sentence at lines 567-568 to make clearer the rationale for focusing on $\delta^{15}N$. And we have added a new supplementary figure S9 showing relationships between $\delta^{18}O_{NO3}$ vs. N\* and fPW.

**Specific comments:**

**R2.5 Materials and methods:**

*Lines 96-107: Is using different types of filters (0.45 $\mu m$ versus 0.22 $\mu m$) affect nutrient concentrations? Was this tested?*

- The use of different filter sizes was an unintended consequence of obtaining samples opportunistically during different expeditions. However, most organisms that could impact nutrient concentrations between the point of collection and subsequent freezing and analysis are larger than 0.45 $\mu m$. We did not perform any specific tests, but there is no indication that any data were impacted by the small difference in filter pore size. We also note that 0.45 $\mu m$ filters are used regularly for nutrient measurements.

**R2.6** *Lines 121-122: Was USGS 32 used to correct $\delta^{15}N$ data? Since its $\delta^{15}N$ is much different from the $\delta^{15}N$ of the - samples, I assume this would be problematic.*

- The reviewer is correct in noting that the USGS 32 standard lies beyond the range of sample $\delta^{15}N_{NO3}$ values. It is used in a 3-point calibration as part of routine operating procedures in the Dalhousie lab. Omitting USGS 32 from the calibration curve had negligible impact (< 0.2 ‰ ) on sample $\delta^{15}N_{NO3}$ values.

**R2.7** *Lines 123-125: Why was $NO_2^-$ not removed? Even small $NO_2^-$ concentrations can affect the $\delta^{15}N$ and $\delta^{18}O$ of $NO_3^-$, especially at low $NO_3^-$ concentrations. What was the lowest $NO_3^-$ concentration for samples analyzed for isotopic composition? Since nitrate concentrations are generally high below the mixed layer, and for most water masses discussed in this manuscript, I don't think this is a major concern.*

- The reviewer is correct in their last point about higher concentrations below the mixed layer in the water masses of concern. The $NO_2^-$ concentrations in this zone were always less than 0.36 $\mu$M, as shown in Figures S3 – S7, which is too low relative to the much higher $NO_3^-$ concentrations (> 6 $\mu$M) to affect the isotopic compositions. We report some isotope data from mixed layer waters with $NO_3^-$ concentrations as low as 0.7 $\mu$M, but since the focus is on waters below the mixed layer, this is not a major concern. We have added text at lines to 158 and 165-166 to make this clearer.

**R2.8** *Another concern is that the "denitrifier" method is sensitive (to some extent) to the $\delta^{18}O$ value of the sample water because of O exchange with water during the conversion of NO3- to N2O. This is an issue if the $\delta^{18}O$ of the samples and standards are drastically different or in polar regions where the $\delta^{18}O$ of water is greatly variable due to mixing between freshwater from rivers and glaciers (with a low $\delta^{18}O$ around -20 ‰) and seawater ($\delta^{18}O$ of about 0‰). Was this taken into account while analyzing their samples? See Kobayashi et al. (2021) for more detail.*

- We did not apply a correction for the $\delta^{18}O$ of seawater. However, the $\delta^{18}O$ in the sample waters range only from -2.2 ‰ to 0.2 ‰ (Lehmann et al. 2019) which would have a minor impact on the $\delta^{18}O_{NO3}$ results. We have added text at lines 151-152 to make this clearer.

**R2.9** *Finally, what was their blank size (i.e., for the bacteria method)?*

- Blank sizes are now stated on lines 158-160 and 166-167.

**R2.10** *Lines 144-145: The authors should make a clearer distinction between regenerated (calculated using Redfield $P:O_2$ ratio and AOU) versus preformed $PO_4^{3-}$ here. Preformed nutrients are those that were present in solution when the parcel of water sank from the surface and are characteristic of different water masses:*
*pre-formed $[PO_4^{3-}]$ = measured $[PO_4^{3-}]$ + regenerated [PO43-]*

- To avoid any confusion, we have restated this as "the ratio of regenerated to *measured* $PO_4^{3}$" and is now reported as $P_{reg/meas}$ throughout the ms.

**R2.11** *Lines 160-161: Explain what cause that kink in the $NO_3^-$ vs $PO_4^{3-}$ relationship at low nutrient concentrations (i.e., nitrate assimilation).*

- We have added explanatory text at line 203, as suggested.

***Results and Discussion:***
**R2.12** *Lines 367-371: I am curious about the isotope effect for nitrate assimilation derived from these relationships and how it compares to previous field studies (e.g., Altabet et al. (2001)?*

- We calculated the apparent fractionation for $NO_3^-$ assimilation for the few stations with a sufficient number of shallow water $\delta^{15}N_{NO3}$ data, based on the slopes of $\delta^{15}N_{NO3}$ versus the negative natural logarithm of the fraction sub-euphotic zone $NO_3^-$ concentrations. The fractionation factors were smaller (0.8 to 3.9 ‰) than the generally assumed 5‰ fractionation for assimilation, but consistent with observations in Granger et al. (2010). However, we choose not to report these results, as they do not address the objectives of the manuscript.

**R2.13** *Lines 415-416: Why isn't the correlation showed for $\delta^{18}O$-$NO_3^-$? Could $\delta^{18}O$ of $NO_3$ also be used as a complementary tool to trace these different water masses? This aspect should be better discussed.*

- With respect to the first point, the existing Figure S8 also includes the $\delta^{18}O_{NO3}$ data with 'r' values in the correlation matrix, but we choose not to report the specific values in this section of the main text, as this would do little to address the main objectives. With regard to discussion of the $\delta^{18}O_{NO3}$ data, please see our response to point R2.4 above.

**R2.14** *Lines 427-429: The authors should explain why no change in water-column $\delta^{15}N$-$NO_3^-$ is expected during sedimentary denitrification (i.e., discuss the suppressed net "community" isotope effect for sedimentary denitrification due to diffusion limitation and complete NO3-consumption in the sediments).*

- We have revised the working on lines 520-522 to explain why sedimentary denitrification has negligible impact on water column $\delta^{15}N_{NO3}$ in deep Baffin Bay waters.

**R2.15** *Line 468: Could a similar equation be derived for $\delta^{18}O$ of $NO_3^-$ as well? However, $\delta^{18}O$ of $NO_3^-$ would not be useful for food-web or paleoceanographic studies as the O atom is not conserved during N incorporation into organic material.*

- The reviewer is correct in their assessment that $\delta^{18}O_{NO3}$ has little utility with respect to foodweb or paleoceanographic applications, in the sense that oxygen is not conserved during incorporation into biological materials. This is our rationale for not showing the relationship between $\delta^{18}O_{NO3}$ and fPW in the main text. We now articulate this more clearly on line 569-568. Note that we have also added the plots of $\delta^{18}O_{NO3}$ vs. N* and fPW as a new supplementary Figure (S9).

**R2.16** *Lines 471-478: This argument needs to be discussed better since as for N*, it is not possible to disentangle different co-occurring processes (nitrification, denitrification, $N_2$ fixation) using $\delta^{15}N$ of $NO_3^-$ data solely. These co-occurring processes were disentangled in the BBW because of the additional insights from $\delta^{18}O$ of $NO_3^-$ data.*

- We now clarify on line 581 that the processes occurring in the deep Baffin Bay water (BBW) would not have been obvious without *paired* N:P and $\delta^{15}N_{NO3}$ data. We have also added the reference to Lehmann et al. (2019), which explore $NO_3$ isotopic ratios in the BBW in greater detail.

**R2.17** *Line 483: This section title is vague. I would rename it "Using our $\delta^{15}N$-$NO_3^-$ relationship to establish a baseline $\delta^{15}N$ for food-web and paleoceanographic studies." This section could also be merged with sections 3.7 and 3.8.*

- This section has been retitled "4.3 Incorporation of $\delta^{15}N_{NO3}$ into "baseline" $\delta^{15}N$ for foodweb and paleoceanographic studies". We prefer to maintain the original structure of sections 3.6, 3.7 and 3.8, which are now re-numbered as sections 4.3, 4.4, and 4.5.

**R2.18** *Line 565: Change to "fraction of Pacific water"*

- We have made this change.

**R2.19** *Table 1. Add number of samples analyzed for each water masses (n).*
*Indicating a range of depths for each water masses would be better than showing the average (given the large standard deviation).*

- We have incorporated both of these suggestions into Table 1.

**R2.20** *Figure 3. Are $\delta^{18}O$ of $NO_3^-$ values shown at about 200 m depth (177) and 500 m depth (ROV5) outliers? It is unclear why there is no corresponding increase in the $\delta^{15}N$ of NO3- at these stations/depths. Were these samples measured in duplicate?*

- These are most likely analytical outliers, especially given the lack of a corresponding increase in $\delta^{15}N_{NO3}$. Unfortunately, neither of these samples was measured in duplicate. In the interests of data transparency, we choose to show these data. However, we have removed the dashed lines connecting these points to the rest of the profile data, which should help to improve figure clarity. We explain this in the figure caption. We have also made the corresponding change to the presentation of $\delta^{18}O_{NO3}$ data in Figures S4 and S6. Note that the treatment of these two data points as analytical outliers also affects the regression statistics for $\delta^{18}O_{NO3}$ vs fPW (now shown in Fig. S9) and the resulting predictions for Atlantic and Pacific end-member $\delta^{18}O_{NO3}$ values (Lines 553-554). These changes are modest and have no impact on our interpretations.

**R2.21** *Figure 5. I think it would make sense to separate the symbols based on depths for this figure (e.g., as in Figure 2: surface waters impacted by nitrate assimilation (open symbols) versus deeper waters (filled symbols)).*

- Respectfully, we disagree with this suggestion to reformat the symbols in Fig. 5 because it would add significant complexity to an already data-rich plot. However, to improve clarity we have rescaled the symbol sizes representing $NO_3^-$ concentrations, which should make it easier for the reader to infer which samples were impacted by $NO_3^-$ utilization.

**2.22** *Figure 7. The R2 as well as p-values should be added.*
- We have added $r^2$ and p-values to the plot, as requested.

---

## Author Response (AR2)

June 26, 2021

Dear Dr. van der Meer,

Thank you for the positive re-review and final acceptance of our manuscript bg-2021-45 "Stable isotope ratios in seawater nitrate reflect the influence of Pacific water along the Northwest Atlantic margin".

We are particularly grateful for the deadline extensions, and for the constructive criticisms throughout the peer-review process.

We have altered the sentence on line 29 per your suggestion in your decision letter. The rest of the ms remains the same as in our resubmission on Jun 16.  We have chosen Figure 7 as the key figure.

I look forward to seeing our paper published in Biogeosciences.

Sincerely,
Owen Sherwood